# Bounded Hyperbolic Tangent: A Stable and Efficient Alternative to Pre-Layer Normalization in Large Language Models

Hoyoon Byun [1]   Youngjun Choi [1]   Taero Kim [1]   Sungrae Park [2]   Kyungwoo Song [1] [*]

## Abstract

Pre-Layer Normalization (Pre-LN) is the de facto choice for large language models (LLMs) and is crucial for stable pretraining and effective transfer learning. However, Pre-LN incurs repeated statistical-computation overhead and remains vulnerable to the curse of depth, where hidden-state magnitudes and variances grow as the number of layers increases, destabilizing training. Efficiency-oriented normalization-free methods such as Dynamic Tanh (DyT) improve throughput but remain fragile at depth. To jointly address stability and efficiency, we propose Bounded Hyperbolic Tanh (BHyT), a drop-in replacement for Pre-LN. BHyT combines a tanh nonlinearity with explicit, data-driven input bounding to keep activations within a non-saturating range. It prevents depth-wise growth in activation magnitude and variance and provides a theoretical stability guarantee. For efficiency, BHyT computes exact statistics once per block and replaces a second normalization with a lightweight variance approximation. Empirically, BHyT demonstrates improved stability and efficiency during pretraining, achieving an average of 1.6% faster training and an average of 1.77% higher token generation throughput compared to RMSNorm, while maintaining strong pretraining-only and post-SFT performance across language understanding and reasoning benchmarks[1].

## 1. Introduction

The remarkable progress of large language models (LLMs) is tightly coupled to the Transformer architecture, in which normalization layers play a central role (Vaswani et al., 2017). In particular, Pre-Layer Normalization (Pre-LN), which applies normalization before the self-attention and MLP sublayers in each Transformer block, has become the de facto standard (Xiong et al., 2020; Radford et al., 2019; Brown et al., 2020; Grattafiori et al., 2024). Many modern LLMs instantiate Pre-LN with RMSNorm (Zhang & Sennrich, 2019), a computationally lighter variant that removes mean centering while retaining per-token scale normalization. By regulating activation scale, these normalization layers stabilize optimization in deep networks, enabling training at scale and strong transfer performance across diverse tasks (Ba et al., 2016; Xiong et al., 2020).

As model depth increases, however, the Pre-LN design exposes an important limitation. Recent analyses identify a *curse of depth*, in which the interaction between residual connections and Pre-LN can drive rapid growth in the magnitude and variance of the hidden state across layers (Sun et al., 2026). This escalation is more consequential than numerical instability alone: it can push the block Jacobian toward the identity, causing deeper layers to behave increasingly like costly identity mappings (Sun et al., 2026). The central challenge is therefore not only to prevent divergence, but also to preserve informative transformations and efficient signal propagation so that each layer contributes meaningfully to learning.

Existing approaches improve either stability or efficiency, but typically not both. Stability-oriented methods strengthen normalization to suppress depth-wise drift. For example, Peri-LN (Kim et al., 2025) applies normalization both before and after each sublayer, reducing variance growth and improving gradient behavior in very deep Transformers. This stability, however, comes at a computational cost: additional normalization operations increase reduction overhead and memory traffic, thereby adding latency. Efficiency-oriented normalization variants reduce part of this cost.

Normalization-free alternatives further reduce overhead, but they do not necessarily control depth-wise residual growth.

---

[*]Co-corresponding author. [1]Department of Statistics and Data Science, Yonsei University, Seoul, South Korea [2]Upstage AI, South Korea. Correspondence to: Kyungwoo Song <kyungwoo.song@yonsei.ac.kr>.

*Proceedings of the 43$^{rd}$ International Conference on Machine Learning*, Seoul, South Korea. PMLR 306, 2026. Copyright 2026 by the author(s).

[1]Code is available at: https://github.com/MLAI-Yonsei/BHyT

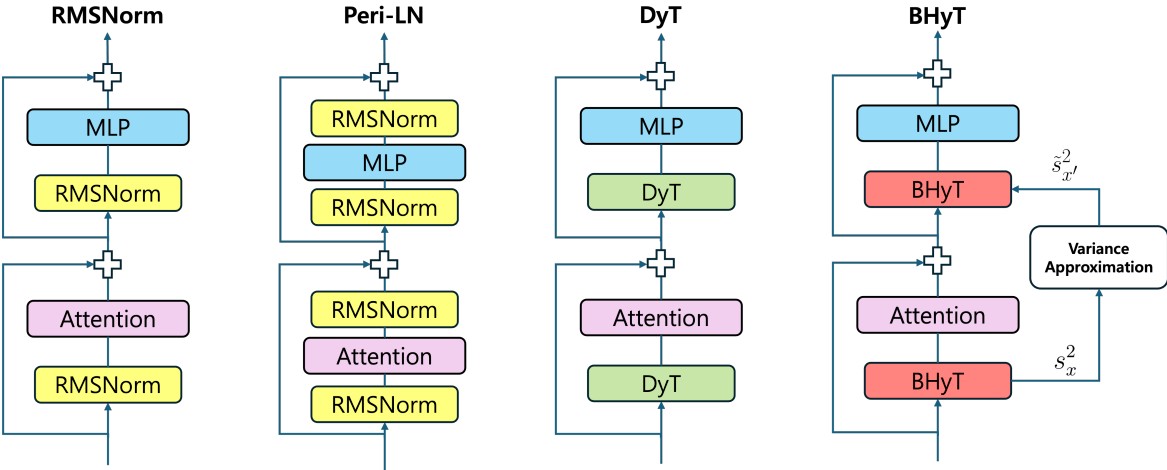

*Figure 1.* Architectural comparison of normalization strategies in Transformer blocks. **RMSNorm** applies normalization before each sublayer to stabilize activations but suffers from variance growth at scale. **Peri-LN** reinforces stability by normalizing both before and after each sublayer. **DyT** replaces normalization with a lightweight scaled tanh nonlinearity with learnable scalar $\alpha$. **BHyT** (ours) combines bounded $\tanh$ with data-driven variance control: it computes input variance once per block, approximates subsequent variance for efficiency, and explicitly constrains activations to a non-saturating range, thereby unifying stability and efficiency.

Dynamic Tanh (DyT) (Zhu et al., 2025) replaces normalization with an element-wise $\tanh(\cdot)$ transformation followed by learnable affine parameters. By eliminating per-token statistic computation, DyT can improve training and inference throughput. However, a bounded element-wise output alone is insufficient to control the residual stream in a pre-normalization residual layout: residual updates can still accumulate across layers, allowing activation magnitude and variance to grow with depth. Moreover, because DyT relies on learned global scaling rather than data-dependent input bounding, large pre-$\tanh$ inputs can enter saturated regimes, weakening forward-path control over depth-wise activation growth. Appendix C.10 illustrates this mechanism using a controlled tresidual-stack analysis.

These limitations reveal a central gap in current normalization and normalization-free methods: stability-oriented approaches often introduce additional computational overhead, whereas more efficient alternatives provide limited control over depth-wise activation growth. A practical replacement for Pre-LN should therefore combine the low overhead of bounded transformations with explicit control over residual-stream scale and variance.

We introduce Bounded Hyperbolic Tanh (BHyT) to address this gap. As shown in Figure 1, BHyT is a drop-in replacement for Pre-LN that couples a $\tanh(\cdot)$ transformation with explicit, data-driven input bounding. By keeping the pre-$\tanh$ argument within a non-saturating range with high probability, BHyT limits depth-wise growth in activation magnitude and variance while maintaining effective forward signal propagation as depth increases.

Beyond this forward-path control, BHyT admits a finite-depth variance propagation bound under explicit hyperparameter conditions. Specifically, for a network of depth $L$, the output variance under BHyT is provably smaller than that under LayerNorm Scaling (LNS) (Sun et al., 2026) under suitable hyperparameter conditions. This result provides theoretical support for BHyT as a mechanism for suppressing depth-wise variance growth, complementing its empirical stability in deep Transformer models.

To reduce overhead, BHyT avoids recomputing the exact variance at both Pre-LN sites in each Transformer block. It computes the input variance once, reuses this statistic, and replaces the second exact variance computation with a lightweight attention-output variance approximation. This block-level design reduces reduction operations and memory movement while preserving the input-bounding mechanism that stabilizes the pre-$\tanh$ argument. As a result, BHyT improves the stability–efficiency trade-off without requiring exact statistic computation at every Pre-LN site.

Empirically, BHyT improves the stability–efficiency trade-off in LLM pretraining across model scales. For Llama-374M and Llama-1B models pretrained on C4 (Raffel et al., 2020) with model-size-proportional token budgets, BHyT suppresses depth-wise activation growth while maintaining strong 3-shot pretraining-only and post-SFT performance. On Llama-1B, BHyT reaches 83.0K training tokens/s and 1,199.9 generation tokens/s, outperforming RMSNorm by 1.63% in training throughput and 1.77% in average generation throughput. We further include a 20B-token comparison between Llama-3B models using BHyT and Peri-LN (Kim et al., 2025), showing that BHyT retains its advantage over a strong stability-oriented baseline at a larger scale, with a full-baseline study on Llama-3B provided in Appendix C.5.

Our contributions are summarized as follows.

- We propose BHyT, a drop-in replacement for normalization at Pre-LN sites that combines a bounded $\tanh(\cdot)$ transformation with data-driven input bounding to control depth-wise activation growth.

- We provide a finite-depth variance analysis showing that BHyT yields a smaller variance-propagation bound than LNS under explicit hyperparameter conditions, and we introduce a block-level variance approximation that reduces repeated exact variance computation.

- We evaluate BHyT on Llama-374M and Llama-1B models pretrained on C4, and further study Llama-3B through a 20B-token comparison against Peri-LN. Across stability, performance, and efficiency evaluations, BHyT improves the stability–efficiency trade-off while maintaining strong downstream performance.

## 2. Related works

### 2.1. Limitations of Pre-layer Normalization

Layer normalization has been a key component in Transformer architectures (Ba et al., 2016), and its Pre-LN variant has become the default in most LLMs (Xiong et al., 2020). The Pre-LN design, which normalizes hidden states before attention and feed-forward layers, improves stability during training. Still, this design is not without drawbacks. From a stability perspective, recent studies have shown that Pre-LN suffers from massive activations as depth increases (Sun et al., 2026; Kim et al., 2025). In this regime, the mean and variance of hidden states grow across layers, destabilizing optimization. From an efficiency perspective, normalization requires computing statistics such as the mean and variance of activations. These reductions introduce latency and memory overhead, which accumulate as depth increases. As a result, Pre-LN can destabilize very deep models and become a bottleneck for training throughput and scalability.

### 2.2. Normalization-based Enhancements

The main bottleneck of normalization-based designs lies in the repeated computation of statistics in every block. A first attempt to reduce this cost was RMSNorm, which discards the mean and normalizes activations only by their root mean square of activations (Zhang & Sennrich, 2019). This removes part of the overhead while keeping activations within a reasonable range. However, per-token variance computation is still required for every layer, and the instability of Pre-LN remains unresolved.

To address this instability, LayerNorm Scaling (LNS) (Sun et al., 2026) introduces a layer index scaling that constrains variance growth across depth. By rescaling activations according to the layer index, it alleviates the massive activation problem and stabilizes optimization in very deep networks.

Although LNS improves stability, its reliance on per-block statistics keeps its cost high in large-scale training. Peri-LN further reinforces stability by applying normalization both before and after each sublayer (Kim et al., 2025). This suppresses variance spikes and improves convergence and downstream accuracy; however, doubling normalization operations adds substantial overhead, making it less suitable for large-scale pretraining.

### 2.3. Normalization-free Transformers

To avoid the overhead of statistical computation altogether, recent work has explored removing normalization. DyT follows this direction by replacing RMSNorm with a bounded activation function, $\tanh(\alpha_{\text{DyT}}x)$, where $\alpha_{\text{DyT}}$ is a learnable scalar (Zhu et al., 2025). This eliminates the need for mean and variance estimation, improving both training and inference throughput. The $\tanh(\cdot)$ ensures that activations remain bounded, and the learnable scalar $\alpha_{\text{DyT}}$ rescales inputs so that the effective range of $\tanh$ is better utilized.

Although DyT improves efficiency, it does not explicitly control how the output mean and variance grow with depth. Moreover, its learnable scalar parameter $\alpha_{\text{DyT}}$ is not specifically designed for stability. It only rescales the inputs indirectly, without actively suppressing variance escalation in deep networks. As a result, its training stability at scale is not explicitly guaranteed and may be more susceptible to saturation-induced vanishing gradient and related instabilities than normalization-based methods.

In contrast to DyT, BHyT provides an explicit stability guarantee by constraining variance growth. To address the computational bottleneck of the required statistical calculations, we introduce a variance approximation mechanism. This design effectively combines robust training stability with the efficiency benefits of normalization-free architectures.

Concurrent work has explored complementary alternatives. Derf (Chen et al., 2025) replaces the $\tanh(\cdot)$ function in DyT with an $\text{erf}(\cdot)$-based activation, whereas SeeD-Norm (Cai et al., 2026) remains within the normalization family by dynamically rescaling RMSNorm based on the current input. Relative to these methods, BHyT is designed as a bounded, data-dependent alternative to Pre-LN that ties the pre-$\tanh$ range directly to per-token statistics while also providing finite-depth variance control.

## 3. Methodology

BHyT serves as a practical replacement for normalization at Pre-LN sites, controlling depth-wise activation growth while preserving the efficiency benefits of bounded transformations. The key idea is to use a $\tanh(\cdot)$ nonlinearity, as in DyT, but to explicitly bound its input using data-dependent statistics rather than relying only on a learnable global scalar

parameter. This input bounding keeps the pre-$\tanh$ argument in a non-saturating range with high probability, thereby helping to limit the magnitude and variance of residual updates across depth. We first derive an ideal form, BHyT$^*$, that computes input statistics exactly, and then introduce the practical BHyT implementation that reduces statistical computation through block-level variance approximation.

### 3.1. Input Bounding for Stable Bounded Transformations

To avoid saturation, BHyT constrains the pre-$\tanh$ input to lie within a predefined interval $[-\lambda, \lambda]$ with high probability. This condition preserves the local sensitivity of the bounded transformation while helping to limit the scale of residual updates. Rather than imposing a deterministic coordinate-wise bound, we derive a probabilistic scaling rule from the mean and variance of the input distribution.

Specifically, we use Chebyshev's inequality to obtain a distribution-agnostic high-probability bound. For any random variable $X$ with a finite mean $\mu$ and variance $s^2$, and for any $\kappa > 1$,

$$\mathbb{P}\left(|X - \mu| \le \kappa s\right) \ge 1 - \kappa^{-2}. \tag{1}$$

This bound implies that most coordinates lie within $\kappa$ standard deviations of the mean, without assuming a particular input distribution.

Motivated by this distribution-agnostic bound, we first define an idealized form, BHyT$^*$, that uses the exact mean and standard deviation of the input. Given $x \in \mathbb{R}^d$ with a mean $\mu_x$ and a standard deviation $s_x$ computed across coordinates, BHyT$^*$ rescales $x$ so that a coordinate of the pre-$\tanh$ input lies in $[-\lambda, \lambda]$ with probability at least $1 - \kappa^{-2}$:

$$\text{BHyT}^*(x) = \gamma \odot \tanh\left(\frac{\lambda}{\kappa s_x + |\mu_x|} x\right). \tag{2}$$

Here, $\gamma \in \mathbb{R}^d$ is a learnable scale parameter, $\odot$ denotes element-wise multiplication, and $\lambda > 0$ specifies the desired pre-$\tanh$ range. For a target probability level $p \in (0, 1)$, we set $\kappa = (1 - p)^{-1/2}$, as formalized in Proposition 3.1. The proof is provided in Appendix B.1.

**Proposition 3.1** (Input scaling bound at the probability level $p$). *Let $x \in \mathbb{R}$ with $\mathbb{E}[x] = \mu_x$ and $\text{Var}(x) = s_x^2$. Given a predefined bound $\lambda > 0$ and target probability $p \in (0, 1)$, if $\alpha = \frac{\lambda}{\kappa s_x + |\mu_x|}$, then $P(|\alpha x| \le \lambda) \ge p$ for any distribution with finite variance, where $\kappa := (1 - p)^{-1/2}$.*

*Remark* 3.2. For $p = 0.99$, we have $\kappa = 10$ and therefore

$$\alpha = \frac{\lambda}{10 s_x + |\mu_x|}.$$

Thus, $\lambda$ controls the target pre-$\tanh$ range, while $\kappa$ determines how conservatively the input is scaled.

### 3.2. Accelerating BHyT with Variance Approximation

The exact form BHyT$^*$ provides a clean probabilistic input bound, but it requires per-instance mean and variance computation at each normalization site. This would weaken the main efficiency benefit of replacing RMSNorm-like normalization with a bounded transformation. We therefore introduce a practical BHyT module that maintains the same bounded $\tanh(\cdot)$ structure but avoids repeated exact variance computation through a block-level variance approximation.

BHyT follows the RMSNorm-style approximation that treats the input as zero-mean and therefore uses variance-based scaling without mean-centering. Within each Transformer block, BHyT computes the input variance exactly once at the first Pre-LN site, reuses this statistic, and approximates the variance needed at the second Pre-LN site after attention. This design reduces repeated reduction operations and memory movement while preserving the input-bounding mechanism that stabilizes the pre-$\tanh$ argument.

#### 3.2.1. BHyT before the Attention Layer

At the first Pre-LN site of a Transformer block, BHyT computes the per-instance variance of the input $x$ across the feature dimension and applies the bounded transformation

$$z_{\text{Attn}} = \text{BHyT}_{\text{Attn}}(x) = \gamma_{\text{Attn}} \odot \tanh(\frac{\lambda_{\text{Attn}}}{\kappa s_x} x) \tag{3}$$

Here, $s_x^2$ denotes the per-instance feature variance of $x$, and $\lambda_{\text{Attn}}$ specifies the target pre-$\tanh$ range at the Pre-LN site before the attention layer. The output $z_{\text{Attn}}$ is then passed to the self-attention layer, producing $h_{\text{Attn}} = \mathcal{A}(z_{\text{Attn}})$.

Equation 3 follows from BHyT$^*$ under the RMSNorm-style zero-mean approximation, which removes the $|\mu_x|$ term from the scaling denominator. For the corresponding zero-mean BHyT transformation, the Jacobian is a diagonally scaled version of the RMSNorm Jacobian, yielding

$$\left\| J^{\text{BHyT}}(x) \right\|_2 \le \frac{\lambda}{\kappa} \left\| J^{\text{RMS}}(x) \right\|_2.$$

The full derivation is provided in Appendix B.2.

#### 3.2.2. BHyT before the MLP Layer

Let $x' = x + h_{\text{Attn}}$ denote the output of the first residual connection and serves as the input to the second Pre-LN site, BHyT$_{\text{MLP}}$. Instead of recomputing the feature variance of $x'$ exactly, BHyT approximates it as

$$\tilde{s}_{x'}^2 = s_x^2 + \tilde{s}_{h_{\text{Attn}}}^2. \tag{4}$$

The first term, $s_x^2$, is reused from the exact computation at the first Pre-LN site. The second term, $\tilde{s}_{h_{\text{Attn}}}^2$, approximates

**Algorithm 1** Transformer Decoder Block with BHyT

**Require:** Token embedding $x \in \mathbb{R}^d$,
**Require:** Hyperparameters $\lambda_{\text{Attn}}, \lambda_{\text{MLP}}, \kappa > 1$, sequence length $T$
**Require:** Learnable scales $\gamma_{\text{Attn}}, \gamma_{\text{MLP}} \in \mathbb{R}^d$
**Require:** Weights $W_V \in \mathbb{R}^{d \times d_V}$, $W_O \in \mathbb{R}^{d_V \times d}$, $W_1 \in \mathbb{R}^{d \times d_m}$, $W_2 \in \mathbb{R}^{d_m \times d}$
**output** $x_{\text{out}} \in \mathbb{R}^d$
1: $s_x^2 \leftarrow \frac{1}{d} \sum_{j=1}^d x_j^2$ ▷ {RMS-style input variance}
2: **parallel do**
3:    **(A) Variance approximation**
4:    $\tilde{s}_{x'}^2 \leftarrow s_x^2 + \frac{1}{Td} \|W_V W_O\|_F^2 \left( \frac{\lambda_{\text{Attn}}}{\kappa} \right)^2$
5:
6:    **(B) Attention path**
7:    $s_x \leftarrow \sqrt{s_x^2}$
8:    $\alpha_{\text{Attn}} \leftarrow \frac{\lambda_{\text{Attn}}}{\kappa s_x}$
9:    $z_{\text{Attn}} \leftarrow \gamma_{\text{Attn}} \odot \tanh(\alpha_{\text{Attn}} x)$ ▷ {BHyT$_{\text{Attn}}$}
10:    $h_{\text{Attn}} \leftarrow \mathcal{A}(z_{\text{Attn}}; W_V, W_O)$ ▷ {Attention sub-layer}
11: **end parallel**
12: $x' \leftarrow x + h_{\text{Attn}}$ ▷ {Residual}
13: $\tilde{s}_{x'} \leftarrow \sqrt{\tilde{s}_{x'}^2}$
14: $\alpha_{\text{MLP}} \leftarrow \frac{\lambda_{\text{MLP}}}{\kappa \tilde{s}_{x'}}$
15: $z_{\text{MLP}} \leftarrow \gamma_{\text{MLP}} \odot \tanh(\alpha_{\text{MLP}} x')$ ▷ {BHyT$_{\text{MLP}}$}
16: $h_{\text{MLP}} \leftarrow \text{MLP}(z_{\text{MLP}}; W_1, W_2)$ ▷ {MLP sublayer}
17: $x_{\text{out}} \leftarrow x' + h_{\text{MLP}}$ ▷ {Residual}
**output** $x_{\text{out}}$

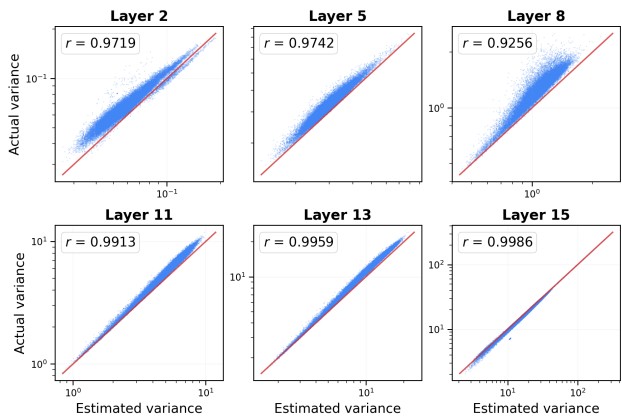

*Figure 2.* Approximated and actual activation variances for the second $\text{BHyT}_{\text{MLP}}$ layer across selected Transformer blocks (layers 2, 5, 8, 11, 13, and 15) in Llama-1B. The variance estimates are evaluated using 100 randomly sampled inputs from the C4 training corpus. The diagonal line represents the ideal $y = x$ reference, and $r$ denotes the Pearson correlation coefficient between the approximated and actual variances. Complete results for all 16 layers are reported in Appendix C.11.

$\frac{1}{d} \sum_{k=1}^d Var((h_{Attn})_k)$, *is approximated as*

$$\tilde{s}_{h_{Attn}}^2 \approx \frac{1}{Td} \|W_V W_O\|_F^2 \cdot \frac{\lambda_{Attn}^2}{\kappa^2}. \qquad (6)$$

The quantity $\tilde{s}_{h_{\text{Attn}}}^2$ is a model-dependent variance estimate rather than an instance-wise statistic computed from the current input. In high-dimensional hidden spaces, activation variance is typically well approximated by its expectation, making this model-level estimate a practical proxy for the instance-wise variance. We empirically verify the quality of this approximation in Figure 2.

Theorem 3.4 shows that the attention-output variance can be approximated using model-level quantities: the attention projection weights, the sequence length, and the BHyT parameters. Because this approximation does not depend on current activations, it can be computed in parallel with the forward pass or updated periodically. Figure 2 shows that the approximate variance closely tracks the empirical variance of intermediate hidden states. Algorithm 1 summarizes this block-level computation, in which the RMS-style input variance $s_x^2$ from the first BHyT site is reused to approximate the second-site variance $\tilde{s}_{x'}^2$. The proof is provided in Appendix B.3.

At inference, $W_V$ and $W_O$ remain fixed, so the model-dependent term can be precomputed or cached with the parameters. In training, the projection weights change over optimization steps; therefore, we update the approximation term periodically rather than recomputing it at every step. The variance approximation avoids adding an additional reduction operation to the critical forward path.

the variance contributed by the attention output under the simplifying assumptions stated in Assumption 3.3. Using $\tilde{s}_{x'}^2$, the MLP-side BHyT is computed as

$$z_{\text{MLP}} = \text{BHyT}_{\text{MLP}}(x') = \gamma_{\text{MLP}} \odot \tanh\left( \frac{\lambda_{\text{MLP}}}{\kappa \tilde{s}_{x'}} x' \right). \quad (5)$$

This avoids a second exact variance computation within the block while preserving the bounded transformation form.

**Assumption 3.3.** For the purpose of deriving a tractable approximation, we consider an idealized attention block in which the attention weights are approximately uniform over a sufficiently long sequence of length $T$ (Ledoux, 2001). We assume that the token-level activations $x \in \mathbb{R}^d$ and the projection matrices $W_V$ and $W_O$ are independent and zero-mean, with Gaussian entries. We further assume that the pre-$\tanh$ argument of BHyT remains in the near-linear regime of $\tanh(\cdot)$, so that the variance of the BHyT output is approximated by $\lambda_{\text{Attn}}^2 / \kappa^2$.

**Theorem 3.4** (Variance approximation of the attention output). *Under Assumption 3.3, let $h_{Attn} \in \mathbb{R}^d$ denote the output vector of the attention layer. The average coordinate-wise variance of this output, denoted by $\tilde{s}_{h_{Attn}}^2 :=*

### 3.3. Depth-wise Variance Propagation and Stability of BHyT

**Theorem 3.5** (Finite-depth variance bound of BHyT). *For a network of depth $L$, if the BHyT hyperparameters satisfy $\lambda/\kappa < 1/\sqrt{L}$, then the output variance under BHyT is strictly smaller than that under LNS (Sun et al., 2026) for every layer $\ell$ up to depth $L$, i.e., for all $1 \leq \ell \leq L$:*

$$\tilde{s}_{x_\ell}^{BHyT} < \tilde{s}_{x_\ell}^{LNS}, \quad \forall \ell \in \{1, \dots, L\}. \tag{7}$$

Theorem 3.5 gives a finite-depth variance bound for BHyT: if $\lambda/\kappa < 1/\sqrt{L}$, then for every layer $1 \leq \ell \leq L$, the variance under BHyT is smaller than the corresponding variance under LNS. For example, choosing $\lambda = 1$ and $\kappa = 10$ satisfies the condition for networks with depth $L < 100$, suggesting that BHyT can achieve a smaller variance bound than LNS for typical model depths. The detailed proof is provided in Appendix B.5.5.

## 4. Experiments & results

We evaluate BHyT along three axes that directly reflect its design goals: **stability**, **performance**, and **efficiency**. For stability, we test whether BHyT suppresses depthwise growth in activation magnitude and variance after pretraining under scaling-law-based token budgets (Hoffmann et al.). For performance, we evaluate both pretraining-only and post-SFT models on downstream benchmarks. For efficiency, we measure pretraining throughput, generation throughput, and the effect of the proposed variance approximation. Our main experiments use Llama-374M and Llama-1B, and we further include a 20B-token comparison between Llama-3B models using BHyT and Peri-LN (Kim et al., 2025). A complementary full-baseline Llama-3B study is reported in Appendix C.5. Together, these experiments assess whether BHyT provides a practical replacement for RMSNorm that improves the stability–efficiency trade-off in LLM training and inference.

### 4.1. Experimental Setup

**Models.** We conduct experiments with two Llama-3.2-style (Grattafiori et al., 2024) model scales: Llama-374M and Llama-1B. We instantiate only the architectures and train all models from scratch. Following Chinchilla-style compute-optimal scaling, we allocate approximately 20 training tokens per parameter (Hoffmann et al.), resulting in 7.5B training tokens for Llama-374M and 20B training tokens for Llama-1B. This setup allows us to evaluate whether BHyT remains effective across model scales under comparable compute-aware pretraining budgets. To further assess larger-scale behavior, we additionally train Llama-3B on 20B tokens and compare BHyT with Peri-LN.

**Baselines.** We compare BHyT against four representative normalization or normalization-replacement methods. RMSNorm (Zhang & Sennrich, 2019) is the default normalization used in Llama-style architectures. LNS (Sun et al., 2026) and Peri-LN (Kim et al., 2025) are designed to improve training stability in deep Transformers. DyT (Zhu et al., 2025) replaces conventional normalization with a $\tanh(\cdot)$-based transformation using a learnable scalar parameter $\alpha_{\text{DyT}}$. A side-by-side formulation and comparison of these methods is provided in Appendix C.7.

**Data and evaluation.** Pretraining is performed on the C4 corpus (Raffel et al., 2020). For supervised fine-tuning (SFT), we use Lima1K (Zhou et al., 2023), a compact and high-quality instruction-tuning dataset. We evaluate all pretraining-only and post-SFT models on eight language understanding and reasoning benchmarks: ARC-C and ARC-E (Clark et al., 2018), PIQA (Bisk et al., 2020), HellaSwag (Zellers et al., 2019), OpenBookQA (OBQA) (Mihaylov et al., 2018), Winogrande (Sakaguchi et al., 2021), MMLU (Hendrycks et al., 2020), and BoolQ (Clark et al., 2019). All downstream tasks are evaluated in the 3-shot setting using lm-evaluation-harness (Gao et al., 2024). We report task-specific accuracy and macro-average accuracy across tasks.

**Implementation.** All pretraining and SFT experiments are implemented using LlamaFactory (Zheng et al., 2024). For downstream evaluation, we use standardized task configurations from lm-evaluation-harness (Gao et al., 2024) without modification. For each method, we select hyperparameters from a shared search range using the Llama-1B 20K-step run, and then use the selected configuration for the full pretraining run at each model's token budget. The detailed search ranges, selected configurations, hardware setup, and parallelism strategy are provided in Appendix C.

### 4.2. Pretraining and Fine-tuning Protocols

We pretrain all methods on C4 using token budgets proportional to model size. Llama-374M is trained on 7.5B tokens, and Llama-1B is trained on 20B tokens. These pretrained checkpoints are used for both pretraining-only and post-SFT evaluations. For pretraining-only evaluation, each method is trained once, and downstream accuracy is averaged over five random 3-shot seeds.

For SFT, we fine-tune each pretrained checkpoint on Lima1K for 15 epochs using five random seeds. Following the protocol used in Peri-LN (Kim et al., 2025), we evaluate each SFT run on CommonSenseQA (Talmor et al., 2019) after each epoch and select the checkpoint with the lowest CommonSenseQA validation loss. Each selected checkpoint is then evaluated on the same 3-shot downstream benchmark suite, using the corresponding seed for 3-shot sampling. We

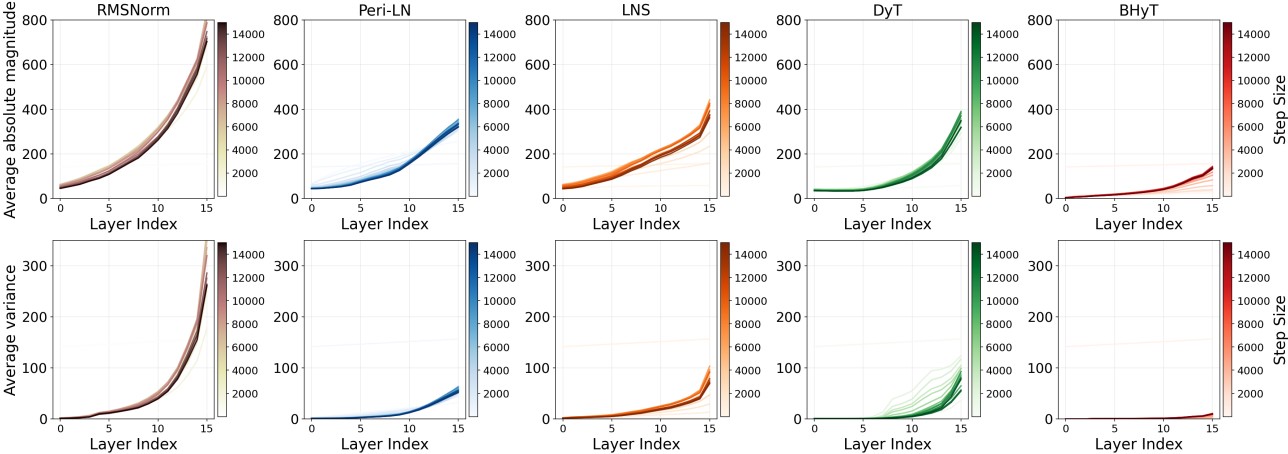

*Figure 3.* Layer-wise activation magnitude and variance for Llama-1B after 20B-token C4 pretraining. Each method is evaluated under its selected best hyperparameter configuration from the shared search. BHyT suppresses depth-wise growth more effectively than RMSNorm and DyT while maintaining controlled activation statistics comparable to stability-oriented normalization methods.

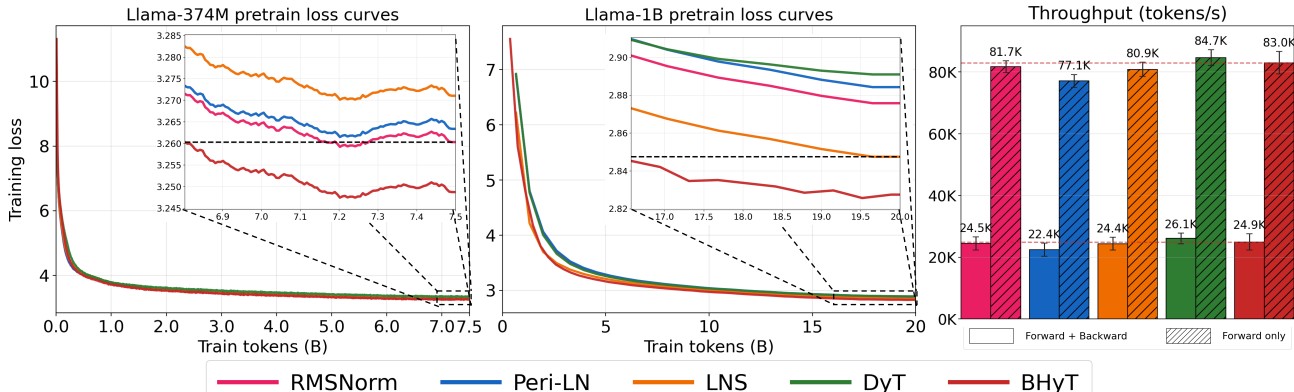

*Figure 4.* Pretraining loss curves and training throughput for Llama-374M and Llama-1B. BHyT achieves stable convergence while providing higher throughput than normalization-based baselines and remaining close to DyT, which avoids statistical computation entirely.

report the average downstream accuracy over the five SFT and evaluation seeds. In addition, we report token-level perplexity (PPL) on the C4 validation set to assess forgetting after SFT and on WikiText-2 (Merity et al., 2016) to assess language modeling generalization beyond the pretraining validation distribution.

### 4.3. Stability Analysis

BHyT stabilizes deep Transformer training by suppressing depth-wise activation growth. Figure 3 reports the layer-wise activation magnitude and variance of Llama-1B after 20B-token C4 pretraining. Unlike a fixed-learning-rate diagnostic, this comparison uses each method's selected best hyperparameter configuration, reflecting the final training setup used for the main experiments. RMSNorm exhibits clear growth in both activation magnitude and variance as depth increases. DyT also shows stronger depth-wise growth than BHyT, despite using its selected configura-

tion; additional learning-rate sensitivity analysis is provided in Appendix C.9. In contrast, BHyT keeps both statistics substantially more controlled across layers. This behavior supports the central motivation of BHyT: replacing RMSNorm with a bounded transformation should not merely remove normalization overhead but should also preserve stable activation dynamics in deep LLMs. Appendix C.10 further analyzes why BHyT suppresses depth-wise growth more effectively than DyT.

### 4.4. Pretraining and SFT Performance

BHyT achieves stable pretraining while improving training throughput across model scales. Figure 4 shows pretraining loss curves for both Llama-374M and Llama-1B, along with training throughput for Llama-1B. BHyT maintains stable convergence on both model scales and achieves higher training throughput than normalization-based baselines, while remaining close to DyT, which avoids statistical computa-

*Table 1.* Pretraining-only evaluation on C4 for Llama-374M and Llama-1B. Training budgets follow approximately 20 tokens per parameter, with Llama-374M trained on 7.5B tokens and Llama-1B on 20B tokens. All downstream tasks are evaluated in the 3-shot setting. BHyT obtains the lowest pretraining loss and the highest average downstream accuracy at both model scales. PPL denotes perplexity.

| Llama-374M | | | | | | | | | | | | |
|---|---|---|---|---|---|---|---|---|---|---|---|---|
| Method | PT Train Loss | PT Eval Loss | PT Eval PPL | ARC-C | ARC-E | PIQA | Hellaswag | OBQA | Winogrande | MMLU | BoolQ | Avg. |
| RMSNorm | 3.532 | 3.216 | 24.916 | $24.13_{\pm0.58}$ | $38.63_{\pm0.45}$ | $\mathbf{66.62}_{\pm0.36}$ | $36.92_{\pm0.15}$ | $26.84_{\pm1.35}$ | $51.51_{\pm0.56}$ | $25.18_{\pm0.02}$ | $50.40_{\pm0.26}$ | $40.03_{\pm0.21}$ |
| Peri-LN | 3.532 | 3.219 | 25.015 | $24.03_{\pm0.43}$ | $37.31_{\pm0.32}$ | $64.91_{\pm0.48}$ | $36.77_{\pm0.23}$ | $27.44_{\pm0.85}$ | $50.70_{\pm0.94}$ | $25.22_{\pm0.01}$ | $\mathbf{51.30}_{\pm0.07}$ | $39.71_{\pm0.18}$ |
| LNS | 3.549 | 3.229 | 25.250 | $23.81_{\pm0.49}$ | $36.32_{\pm0.50}$ | $66.12_{\pm0.33}$ | $36.31_{\pm0.12}$ | $26.72_{\pm0.82}$ | $49.20_{\pm1.13}$ | $26.06_{\pm0.01}$ | $50.24_{\pm0.31}$ | $39.35_{\pm0.24}$ |
| DyT | 3.614 | 3.299 | 27.085 | $24.27_{\pm0.65}$ | $36.66_{\pm0.56}$ | $63.59_{\pm0.35}$ | $34.19_{\pm0.22}$ | $27.64_{\pm0.57}$ | $50.24_{\pm0.70}$ | $25.96_{\pm0.02}$ | $50.42_{\pm0.22}$ | $39.12_{\pm0.18}$ |
| BHyT | **3.519** | **3.207** | 24.714 | $\mathbf{24.40}_{\pm0.35}$ | $\mathbf{38.68}_{\pm0.29}$ | $65.80_{\pm0.42}$ | $\mathbf{37.31}_{\pm0.09}$ | $\mathbf{27.88}_{\pm0.52}$ | $\mathbf{51.71}_{\pm0.56}$ | $\mathbf{26.19}_{\pm0.01}$ | $50.53_{\pm0.63}$ | $\mathbf{40.31}_{\pm0.11}$ |

| Llama-1B | | | | | | | | | | | | |
|---|---|---|---|---|---|---|---|---|---|---|---|---|
| Method | PT Train Loss | PT Eval Loss | PT Eval PPL | ARC-C | ARC-E | PIQA | Hellaswag | OBQA | Winogrande | MMLU | BoolQ | Avg. |
| RMSNorm | 2.876 | 2.877 | 17.753 | $24.33_{\pm0.22}$ | $39.43_{\pm0.47}$ | $68.16_{\pm0.54}$ | $44.83_{\pm0.15}$ | $28.32_{\pm0.72}$ | $50.29_{\pm0.93}$ | $26.55_{\pm0.03}$ | $53.62_{\pm0.21}$ | $41.94_{\pm0.15}$ |
| Peri-LN | 2.884 | 2.884 | 17.892 | $23.17_{\pm0.63}$ | $39.45_{\pm0.50}$ | $68.07_{\pm0.50}$ | $44.69_{\pm0.19}$ | $29.40_{\pm0.63}$ | $52.15_{\pm0.17}$ | $26.99_{\pm0.01}$ | $54.51_{\pm0.71}$ | $42.31_{\pm0.22}$ |
| LNS | 2.848 | 2.848 | 17.259 | $24.15_{\pm0.80}$ | $38.86_{\pm0.17}$ | $68.70_{\pm0.41}$ | $46.52_{\pm0.14}$ | $\mathbf{30.56}_{\pm0.96}$ | $50.80_{\pm0.65}$ | $26.30_{\pm0.02}$ | $\mathbf{56.90}_{\pm0.26}$ | $42.85_{\pm0.11}$ |
| DyT | 2.891 | 2.894 | 18.065 | $\mathbf{24.50}_{\pm0.56}$ | $\mathbf{43.03}_{\pm0.33}$ | $68.74_{\pm0.49}$ | $43.55_{\pm0.13}$ | $28.96_{\pm1.00}$ | $\mathbf{52.41}_{\pm0.38}$ | $\mathbf{27.24}_{\pm0.03}$ | $53.30_{\pm0.71}$ | $42.71_{\pm0.21}$ |
| BHyT | **2.828** | **2.802** | **16.470** | $24.28_{\pm0.43}$ | $40.94_{\pm0.38}$ | $\mathbf{69.06}_{\pm0.36}$ | $\mathbf{48.97}_{\pm0.03}$ | $30.04_{\pm1.13}$ | $51.29_{\pm0.65}$ | $27.01_{\pm0.06}$ | $55.74_{\pm0.43}$ | $\mathbf{43.42}_{\pm0.25}$ |

*Table 2.* Post-SFT evaluation for Llama-374M and Llama-1B after C4 pretraining with model-size-proportional token budgets. Each model is fine-tuned on Lima1K for 15 epochs, and the checkpoint with the lowest CommonSenseQA validation loss is selected. C4 and WikiText-2 token-level PPL are reported to assess post-SFT forgetting and language modeling generalization on a separate corpus, respectively. All downstream tasks are evaluated in the 3-shot setting. BHyT attains the best results in C4 PPL, WikiText-2 PPL, and average downstream accuracy at both model scales.

| Llama-374M | | | | | | | | | | | |
|---|---|---|---|---|---|---|---|---|---|---|---|
| Method | C4 PPL | WikiText-2 PPL | ARC-C | ARC-E | PIQA | Hellaswag | OBQA | Winogrande | MMLU | BoolQ | Avg. |
| RMSNorm | $27.90_{\pm0.03}$ | $39.45_{\pm0.08}$ | $24.45_{\pm0.79}$ | $\mathbf{43.16}_{\pm0.44}$ | $\mathbf{67.57}_{\pm0.43}$ | $\mathbf{37.74}_{\pm0.15}$ | $28.48_{\pm0.57}$ | $50.66_{\pm0.50}$ | $24.83_{\pm0.08}$ | $51.50_{\pm0.74}$ | $41.02_{\pm0.22}$ |
| Peri-LN | $28.20_{\pm0.04}$ | $38.62_{\pm0.09}$ | $24.22_{\pm0.41}$ | $41.98_{\pm0.72}$ | $66.82_{\pm0.30}$ | $37.51_{\pm0.24}$ | $27.96_{\pm0.50}$ | $50.26_{\pm0.51}$ | $24.44_{\pm0.06}$ | $51.84_{\pm0.45}$ | $40.61_{\pm0.20}$ |
| LNS | $28.39_{\pm0.03}$ | $39.47_{\pm0.06}$ | $\mathbf{24.91}_{\pm0.26}$ | $41.28_{\pm0.20}$ | $67.51_{\pm0.33}$ | $37.46_{\pm0.16}$ | $28.32_{\pm0.68}$ | $48.49_{\pm0.43}$ | $26.65_{\pm0.04}$ | $51.34_{\pm0.41}$ | $40.79_{\pm0.09}$ |
| DyT | $31.12_{\pm0.02}$ | $44.67_{\pm0.07}$ | $24.85_{\pm0.34}$ | $41.20_{\pm0.28}$ | $65.06_{\pm0.70}$ | $35.11_{\pm0.18}$ | $28.44_{\pm1.15}$ | $\mathbf{51.32}_{\pm0.59}$ | $\mathbf{26.83}_{\pm0.07}$ | $50.55_{\pm0.34}$ | $40.42_{\pm0.20}$ |
| BHyT | $\mathbf{27.40}_{\pm0.13}$ | $\mathbf{38.39}_{\pm0.39}$ | $\mathbf{25.44}_{\pm0.27}$ | $42.12_{\pm0.48}$ | $66.18_{\pm0.21}$ | $37.31_{\pm0.11}$ | $\mathbf{29.08}_{\pm0.45}$ | $51.21_{\pm0.91}$ | $26.23_{\pm0.12}$ | $\mathbf{52.02}_{\pm0.40}$ | $\mathbf{41.20}_{\pm0.17}$ |

| Llama-1B | | | | | | | | | | | |
|---|---|---|---|---|---|---|---|---|---|---|---|
| Method | C4 PPL | WikiText-2 PPL | ARC-C | ARC-E | PIQA | Hellaswag | OBQA | Winogrande | MMLU | BoolQ | Avg. |
| RMSNorm | $20.34_{\pm0.01}$ | $25.99_{\pm0.01}$ | $\mathbf{27.34}_{\pm0.50}$ | $49.88_{\pm0.40}$ | $71.16_{\pm0.63}$ | $47.93_{\pm0.17}$ | $30.76_{\pm0.92}$ | $51.90_{\pm0.87}$ | $26.80_{\pm0.07}$ | $50.27_{\pm0.74}$ | $44.51_{\pm0.23}$ |
| Peri-LN | $21.33_{\pm0.75}$ | $26.88_{\pm0.97}$ | $25.87_{\pm0.49}$ | $48.93_{\pm0.32}$ | $70.17_{\pm0.29}$ | $47.61_{\pm0.70}$ | $31.76_{\pm1.21}$ | $52.17_{\pm0.57}$ | $25.51_{\pm0.10}$ | $48.79_{\pm0.25}$ | $43.85_{\pm0.12}$ |
| LNS | $19.89_{\pm0.21}$ | $25.99_{\pm0.35}$ | $26.52_{\pm0.28}$ | $48.90_{\pm0.48}$ | $70.74_{\pm0.57}$ | $50.11_{\pm0.21}$ | $31.04_{\pm1.62}$ | $51.76_{\pm0.68}$ | $24.57_{\pm0.09}$ | $54.02_{\pm0.36}$ | $44.71_{\pm0.19}$ |
| DyT | $20.27_{\pm0.01}$ | $24.61_{\pm0.01}$ | $25.70_{\pm0.69}$ | $48.99_{\pm0.35}$ | $70.17_{\pm0.45}$ | $44.92_{\pm0.19}$ | $31.36_{\pm1.35}$ | $\mathbf{52.68}_{\pm0.68}$ | $27.24_{\pm0.07}$ | $50.29_{\pm0.65}$ | $43.92_{\pm0.29}$ |
| BHyT | $\mathbf{19.37}_{\pm0.15}$ | $\mathbf{23.84}_{\pm0.29}$ | $27.18_{\pm0.69}$ | $\mathbf{50.35}_{\pm0.67}$ | $\mathbf{72.29}_{\pm0.23}$ | $\mathbf{50.88}_{\pm0.71}$ | $\mathbf{32.16}_{\pm1.11}$ | $51.07_{\pm0.94}$ | $\mathbf{27.54}_{\pm0.07}$ | $\mathbf{57.05}_{\pm0.91}$ | $\mathbf{46.07}_{\pm0.16}$ |

*Table 3.* Performance of Llama-3B pretrained on 20B tokens, evaluated before and after supervised fine-tuning (SFT). Post-SFT C4 and WikiText-2 PPL measure forgetting after SFT and language modeling generalization on a separate corpus, respectively. BHyT consistently outperforms the stability-oriented baseline, Peri-LN, achieving lower losses, lower PPL, and higher downstream accuracy.

| Llama-3B (Pretrained on 20B tokens only) | | | | | | | | | | | |
|---|---|---|---|---|---|---|---|---|---|---|---|
| Method | PT Train Loss | PT Eval Loss | ARC-C | ARC-E | PIQA | Hellaswag | OBQA | Winograde | MMLU | BoolQ | Avg. |
| Peri-LN | 2.811 | 2.812 | $23.29_{\pm0.56}$ | $40.90_{\pm0.34}$ | $68.86_{\pm0.67}$ | $47.54_{\pm0.13}$ | $30.36_{\pm0.67}$ | $51.51_{\pm0.22}$ | $27.02_{\pm0.01}$ | $47.20_{\pm0.66}$ | $42.08_{\pm0.25}$ |
| BHyT | **2.756** | **2.760** | $\mathbf{25.03}_{\pm0.48}$ | $\mathbf{45.97}_{\pm0.58}$ | $\mathbf{70.18}_{\pm0.08}$ | $\mathbf{51.02}_{\pm0.20}$ | $\mathbf{31.12}_{\pm1.26}$ | $\mathbf{52.14}_{\pm0.29}$ | $\mathbf{27.50}_{\pm0.02}$ | $\mathbf{55.88}_{\pm0.34}$ | $\mathbf{44.86}_{\pm0.11}$ |

| Llama-3B (Pretrained on 20B tokens & SFT) | | | | | | | | | | | |
|---|---|---|---|---|---|---|---|---|---|---|---|
| Method | C4 PPL | WikiText-2 PPL | ARC-C | ARC-E | PIQA | Hellaswag | OBQA | Winograde | MMLU | BoolQ | Avg. |
| Peri-LN | $18.72_{\pm0.06}$ | $23.04_{\pm0.03}$ | $27.51_{\pm0.60}$ | $51.08_{\pm0.50}$ | $71.76_{\pm0.48}$ | $50.10_{\pm0.18}$ | $32.16_{\pm0.53}$ | $52.23_{\pm0.51}$ | $26.54_{\pm0.25}$ | $42.56_{\pm1.19}$ | $44.24_{\pm0.20}$ |
| BHyT | $\mathbf{17.43}_{\pm0.15}$ | $\mathbf{21.33}_{\pm0.20}$ | $\mathbf{28.21}_{\pm0.53}$ | $\mathbf{54.62}_{\pm0.4}$ | $\mathbf{73.46}_{\pm0.43}$ | $\mathbf{53.85}_{\pm0.50}$ | $\mathbf{32.36}_{\pm0.69}$ | $\mathbf{53.12}_{\pm0.40}$ | $\mathbf{26.79}_{\pm0.23}$ | $\mathbf{49.16}_{\pm1.36}$ | $\mathbf{46.45}_{\pm0.28}$ |

tion entirely.

BHyT's controlled activation dynamics translate into strong pretraining-only performance. Table 1 reports pretraining-only results for Llama-374M and Llama-1B under model-size-proportional C4 token budgets. Across both model scales, BHyT achieves the lowest PT train loss, PT evaluation loss, and perplexity. It also obtains the highest average downstream accuracy, outperforming the second-best method by 0.3 points on Llama-374M and 0.6 points on Llama-1B. These results indicate that BHyT's stability advantage is not only visible in internal activation statistics but

*Table 4.* Ablation of the proposed variance approximation. $\text{BHyT}^*$ denotes BHyT with exact variance computation, while BHyT uses the proposed approximation. This ablation follows the short-budget pretraining setting; detailed setup information is provided in Appendix C.6.

| Llama-1B | Variance Approx. | PT Train Loss | PT Eval Loss | PT Eval PPL | Train steps per sec. | ARC-E | PIQA | Hellaswag | OBQA | Winogrande | MMLU | BoolQ | Avg. |
|---|---|---|---|---|---|---|---|---|---|---|---|---|---|
| RMSNorm | X | 3.281 | 3.272 | 26.353 | 0.346 | $30.97_{\pm0.20}$ | $62.89_{\pm0.74}$ | $32.77_{\pm0.09}$ | $32.32_{\pm0.94}$ | $50.54_{\pm0.48}$ | $\mathbf{25.70}_{\pm0.01}$ | $56.26_{\pm0.44}$ | $41.64_{\pm0.10}$ |
| BHyT* | X | **3.266** | **3.254** | **25.885** | 0.335 | $\mathbf{39.82}_{\pm0.52}$ | $\mathbf{64.86}_{\pm0.50}$ | $\mathbf{35.67}_{\pm0.15}$ | $27.68_{\pm0.93}$ | $50.40_{\pm0.80}$ | $25.19_{\pm0.00}$ | $55.34_{\pm0.67}$ | $\mathbf{42.71}_{\pm0.15}$ |
| RMSNorm-Approx | O | *3.293* | *3.284* | *26.672* | *0.381* | $31.15_{\pm0.44}$ | $61.98_{\pm0.57}$ | $31.93_{\pm0.04}$ | $33.08_{\pm0.04}$ | $\mathbf{50.67}_{\pm0.94}$ | $25.16_{\pm0.03}$ | $46.32_{\pm0.26}$ | $40.04_{\pm0.09}$ |
| BHyT | O | *3.268* | **3.254** | *25.908* | ***0.385*** | $30.50_{\pm0.16}$ | $62.42_{\pm0.41}$ | $32.11_{\pm0.10}$ | $\mathbf{33.88}_{\pm0.73}$ | $50.15_{\pm0.32}$ | $25.19_{\pm0.02}$ | $\mathbf{61.86}_{\pm0.17}$ | $42.30_{\pm0.13}$ |

*Table 5.* Generation throughput for Llama-1B. BHyT is faster than normalization-based baselines, while DyT remains the fastest statistics-free baseline.

| Throughput (tokens/s) | Max new token length | |
|---|---|---|
| | 128 | 512 |
| RMSNorm | $1152.2_{\pm20.9}$ (-2.1%) | $1181.1_{\pm2.9}$ (-1.6%) |
| Peri-LN | $957.5_{\pm4.3}$ (-18.6%) | $984.4_{\pm6.9}$ (-18.0%) |
| LNS | $1095.1_{\pm39.9}$ (-6.9%) | $1150.7_{\pm11.7}$ (-4.1%) |
| DyT | $1363.9_{\pm9.3}$ (+15.9%) | $1352.0_{\pm11.9}$ (+12.7%) |
| BHyT | $1176.7_{\pm39.1}$ | $1199.9_{\pm3.0}$ |

also translates into stronger downstream performance after pretraining.

BHyT remains effective after SFT. Table 2 reports post-SFT results on Llama-374M and Llama-1B. Besides downstream 3-shot accuracy, we report PPL on C4 to assess forgetting after SFT and on WikiText-2 to assess language modeling generalization beyond the pretraining distribution. At both model scales, BHyT achieves the lowest C4 PPL, the lowest WikiText-2 PPL, and the highest average accuracy. It outperforms the second-best method in average downstream accuracy by 0.2 points on Llama-374M and 1.4 points on Llama-1B. These results suggest that BHyT's pretraining stability enables reliable adaptation while preserving language modeling ability after SFT.

We further evaluate BHyT at the larger Llama-3B scale under a 20B-token pretraining budget. Table 3 compares BHyT with Peri-LN, the strongest stability-oriented baseline, before and after SFT. In the pretraining-only setting, BHyT achieves lower PT train and evaluation losses and outperforms Peri-LN by 3.4 percentage points in average downstream accuracy. After SFT, BHyT achieves lower PPL on both C4 and WikiText-2 and outperforms Peri-LN by 2.5 percentage points in average downstream accuracy. These results suggest that BHyT's stability advantage persists at a larger model scale under longer pretraining. A short-budget full-baseline comparison at the Llama-3B scale is provided in Appendix C.5.

### 4.5. Efficiency Analysis

BHyT also improves generation throughput compared with normalization-based baselines. Table 5 reports single-GPU generation throughput for Llama-1B using an input length of 512 and a maximum new token length of 512; Appendix C.12 reports additional output lengths and measurement details. BHyT achieves 1,199.9 tokens/s, exceeding RMSNorm, LNS, and Peri-LN by 1.6%, 4.1%, and 18.0%, respectively. DyT remains faster at 1,352.0 tokens/s because it avoids statistical computation entirely. Thus, BHyT offers a practical trade-off: it reduces RMSNorm-like normalization overhead while retaining stronger depth-wise stability than statistics-free DyT.

### 4.6. Variance Approximation Ablation

The variance approximation is essential for making BHyT practically efficient. Table 4 isolates the effect of replacing exact variance computation with the proposed approximation in the short-budget pretraining setting. We provide the detailed setup in Appendix C.6 and focus here on the relative computational and performance trends. For RMSNorm, the approximation improves training throughput from 0.346 to 0.381 steps/s, corresponding to a 10.1% speedup. For BHyT, the approximation improves throughput from 0.335 steps/s for $\text{BHyT}^*$ to 0.385 steps/s, corresponding to a 14.9% speedup. At the same time, BHyT preserves nearly identical PT evaluation loss and perplexity relative to $\text{BHyT}^*$, with only minor changes in downstream accuracy. These results show that the proposed approximation reduces the computational cost of BHyT without materially degrading convergence or downstream performance, making it a practical replacement for RMSNorm.

## 5. Conclusion

We propose Bounded Hyperbolic Tanh (BHyT) to balance stability and efficiency in deep LLMs. By combining explicit probabilistic bounding with a lightweight variance approximation, BHyT mitigates the "curse of depth" without repeated normalization overhead. Theoretical analysis shows that BHyT provides finite-depth variance control, while empirical results demonstrate controlled activation growth, improved throughput compared with normalization-based baselines, and strong pretraining-only and post-SFT performance from 374M to 3B-scale models. BHyT therefore offers a practical path to deeper and more efficient foundation models. Overall, our results suggest that scalable training can improve efficiency without sacrificing stability.

## Acknowledgements

This work was supported by the National Research Foundation of Korea(NRF) grant funded by the Korea government(MSIT)(RS-2024-00457216), also supported by Institute of Information & communications Technology Planning & Evaluation(IITP) under the Leading Generative AI Human Resources Development(IITP-2026-RS-2026-25544647) grant funded by the Korea government(MSIT)

## Impact Statement

This paper presents work whose goal is to advance the field of machine learning by improving the stability and efficiency of large language model training. BHyT may reduce computational overhead in pretraining and inference, potentially lowering the resource cost of developing and deploying language models. The method itself does not introduce a new dataset, user-facing application, or decision-making system. We therefore expect its broader societal implications to be primarily those associated with advances in efficient and scalable machine learning.

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

# Appendix

## A. Statement on the Use of Large Language Models

LLMs were employed solely as a writing assistant to perform limited tasks such as grammar checking and improving readability. The core scientific aspects of this work, including the conception of ideas, development of methodology, theoretical and experimental studies, and drafting of the manuscript, were carried out entirely by the authors without contributions from the LLMs.

## B. Proofs of the Theorems

### B.1. Proof of Proposition 3.1

**Proposition B.1** (Input scaling bound at the probability level $p$). *Let $x \in \mathbb{R}$ with $\mathbb{E}[x] = \mu_x$ and $\mathrm{Var}(x) = s_x^2$. Given a predefined bound $\lambda > 0$ and target probability $p \in (0,1)$, if $\alpha = \frac{\lambda}{\kappa s_x + |\mu_x|}$, then $P(|\alpha x| \leq \lambda) \geq p$ for any distribution with finite variance, where $\kappa := (1-p)^{-1/2}$.*

*Proof.* We aim to guarantee that the scaled variable $\alpha x$ lies within the predefined bound $[-\lambda, \lambda]$ with probability at least $p$, i.e.,

$$P(|\alpha x| \leq \lambda) \geq p.$$

This condition is equivalent to

$$P\left(|x| \leq \frac{\lambda}{|\alpha|}\right) \geq p.$$

By the triangle inequality, we have

$$|x| = |x - \mu_x + \mu_x| \leq |x - \mu_x| + |\mu_x|.$$

Therefore, the event $\{x \mid |x - \mu_x| \leq \kappa s_x\}$ implies $\{x \mid |x| \leq \kappa s_x + |\mu_x|\}$, and hence

$$\{x \mid |x - \mu_x| \leq \kappa s_x\} \subseteq \{x \mid |x| \leq \kappa s_x + |\mu_x|\}.$$

If we choose $\kappa$ such that

$$\kappa s_x + |\mu_x| \leq \frac{\lambda}{|\alpha|},$$

then, by the monotonicity of probability,

$$P\left(|x| \leq \frac{\lambda}{|\alpha|}\right) \geq P(|x - \mu_x| \leq \kappa s_x).$$

Applying Chebyshev's inequality,

$$P(|x - \mu_x| \leq \kappa s_x) \geq 1 - \frac{1}{\kappa^2}.$$

Setting $\kappa := (1-p)^{-1/2}$ yields

$$P\left(|x| \leq \frac{\lambda}{|\alpha|}\right) \geq p.$$

Solving $\kappa s_x + |\mu_x| = \lambda/|\alpha|$ for $\alpha$ gives

$$\alpha = \frac{\lambda}{\kappa s_x + |\mu_x|}.$$

$\square$

## B.2. Deterministic Jacobian bound relative to RMSNorm

We now make explicit the deterministic Jacobian comparison between zero mean BHyT and the corresponding RMSNorm map. Let

$$u(x) := \frac{x}{s_x}, \qquad s_x := \sqrt{\frac{1}{d}\sum_{j=1}^{d} x_j^2},$$

and consider

$$\text{RMSNorm}(x) = \gamma \odot u(x), \qquad \text{BHyT}(x) = \gamma \odot \tanh\left(\frac{\lambda}{\kappa}u(x)\right).$$

Let $J_u(x)$ denote the Jacobian of $u(x)$. By direct differentiation, the Jacobian of RMSNorm is

$$J^{\text{RMS}}(x) = \text{diag}(\gamma)J_u(x).$$

Applying the chain rule to BHyT gives

$$J^{\text{BHyT}}(x) = \text{diag}\left(\gamma \odot \text{sech}^2\left(\frac{\lambda}{\kappa}u(x)\right)\right)\frac{\lambda}{\kappa}J_u(x).$$

Because diagonal matrices commute, we can rewrite this as

$$J^{\text{BHyT}}(x) = D(x)\,\text{diag}(\gamma)J_u(x) = D(x)J^{\text{RMS}}(x),$$

where $D(x) = \text{diag}(d_1(x), \ldots, d_d(x))$ with diagonal entries

$$d_i(x) := \frac{\lambda}{\kappa}\text{sech}^2\left(\frac{\lambda}{\kappa}u_i(x)\right).$$

Since $\text{sech}^2(t) \leq 1$ for all $t \in \mathbb{R}$, every diagonal entry satisfies $d_i(x) \leq \lambda/\kappa$. Therefore,

$$\|D(x)\|_2 = \max_{1 \leq i \leq d} d_i(x) \leq \frac{\lambda}{\kappa}.$$

Using submultiplicativity of the spectral norm, we obtain the deterministic bound

$$\left\|J^{\text{BHyT}}(x)\right\|_2 \leq \|D(x)\|_2 \left\|J^{\text{RMS}}(x)\right\|_2 \leq \frac{\lambda}{\kappa}\left\|J^{\text{RMS}}(x)\right\|_2.$$

This inequality holds for every input $x$ and does not require any distributional assumption. In particular, with the default choice $(\lambda, \kappa) = (1, 10)$, the local Jacobian operator norm of BHyT is at most $0.1$ times that of the corresponding RMSNorm map.

## B.3. Proof of Theorem 3.4

**Theorem 3.4** (Variance approximation of the attention output). *Under Assumption 3.3, let $h_{Attn} \in \mathbb{R}^d$ denote the output vector of the attention layer. The average coordinate-wise variance of this output, denoted by $\tilde{s}_{h_{Attn}}^2 := \frac{1}{d}\sum_{k=1}^{d} Var((h_{Attn})_k)$, is approximated as*

$$\tilde{s}_{h_{Attn}}^2 \approx \frac{1}{Td}\|W_V W_O\|_F^2 \cdot \frac{\lambda_{Attn}^2}{\kappa^2}. \tag{6}$$

*Proof.* We aim to estimate the variance of the attention output to analyze signal propagation without computing the actual forward pass. Under Assumption 3.3, for a large sequence length $T$, the attention weights are approximately uniform. Consequently, the output vector $h_{Attn}$ (viewed as a row vector) can be approximated as the average of the projected value vectors:

$$h_{\text{Attn}} \approx \frac{1}{T}\sum_{t=1}^{T} z_t W_V W_O,$$

where $z_t$ is the activation of the $t$-th token after BHyT, and we let $W = W_V W_O$. Since the inputs and weights are zero-mean, the output $h_{\text{Attn}}$ also has a zero-mean. Therefore, the variance of the $k$-th coordinate is $\text{Var}((h_{\text{Attn}})_k) = \mathbb{E}[(h_{\text{Attn}})_k^2]$. Substituting this into the definition of $\tilde{s}^2_{h_{\text{Attn}}}$, we can express the average variance using the trace of the covariance matrix:

$$\tilde{s}^2_{h_{\text{Attn}}} = \frac{1}{d} \sum_{k=1}^{d} \mathbb{E}[(h_{\text{Attn}})_k^2] = \frac{1}{d} \mathbb{E}[\|h_{\text{Attn}}\|_2^2] = \frac{1}{d} \text{Tr}(\text{Cov}(h_{\text{Attn}})).$$

Since the token activations $z_t$ are assumed to be independent and identically distributed, the covariance of $h_{\text{Attn}}$ is

$$\text{Cov}(h_{\text{Attn}}) \approx \text{Cov}\left(\frac{1}{T} \sum_{t=1}^{T} z_t W\right) = \frac{1}{T^2} \sum_{t=1}^{T} W^\top \text{Cov}(z_t) W.$$

According to Assumption 3.3, the input elements are i.i.d., and BHyT operates element-wise. Thus, the output coordinates of $z_t$ remain uncorrelated, leading to a diagonal covariance matrix. With the sample variance is approximately $\lambda^2/\kappa^2$, we have $\text{Cov}(z_t) \approx \frac{\lambda^2_{\text{Attn}}}{\kappa^2} I_d$. Substituting this into the trace expression yields:

$$\tilde{s}^2_{h_{\text{Attn}}} \approx \frac{1}{d} \text{Tr}\left(\frac{1}{T^2} \sum_{t=1}^{T} W^\top \left(\frac{\lambda^2_{\text{Attn}}}{\kappa^2} I_d\right) W\right) = \frac{1}{d} \text{Tr}\left(\frac{1}{T} \frac{\lambda^2_{\text{Attn}}}{\kappa^2} W^\top W\right).$$

Finally, using the identity $\text{Tr}(W^\top W) = \|W\|_F^2$, we obtain the stated approximation:

$$\tilde{s}^2_{h_{\text{Attn}}} \approx \frac{1}{Td} \|W_V W_O\|_F^2 \frac{\lambda^2_{\text{Attn}}}{\kappa^2}.$$

$\square$

Strictly speaking, the normalization by $s_x$ introduces weak dependencies between feature dimensions. However, for high-dimensional vectors (large $d$), the impact of these dependencies on the covariance structure is negligible ($O(1/d)$), allowing us to approximate $\text{Cov}(z_t)$ as a diagonal matrix.

## B.4. Approximation of the variance of the MLP layer output

**Theorem B.2** (Variance approximation of MLP layer output). *Let $z \in \mathbb{R}^d$ be the input to the MLP sublayer with sample variance $s_z^2$, and let $h_{MLP} \in \mathbb{R}^d$ be the output. The MLP consists of two linear transformations parameterized by $W_1 \in \mathbb{R}^{d \times d_m}$ and $W_2 \in \mathbb{R}^{d_m \times d}$, separated by an element-wise activation function $\phi(\cdot)$. We assume that the activation function scales the covariance of its input by a factor $\tau$ (i.e., $\text{Cov}(\phi(u)) \approx \tau \text{Cov}(u)$). The average sample variance of the MLP output, denoted by $\tilde{s}^2_{h_{MLP}}$, is approximated as:*

$$\tilde{s}^2_{h_{MLP}} \approx \tau \frac{s_z^2}{d} \|W_1 W_2\|_F^2.$$

*Proof.* We analyze the propagation of variance through the MLP block defined by $h_{\text{MLP}} = \phi(zW_1)W_2$. Let $u = zW_1$ be the pre-activation vector and $v = \phi(u)$ be the post-activation vector. Similar to the definition in Theorem 3.4, the average sample variance of the output is given by the trace of its covariance matrix:

$$\tilde{s}^2_{h_{\text{MLP}}} = \frac{1}{d} \text{Tr}(\text{Cov}(h_{\text{MLP}})).$$

We expand the covariance of the output with respect to the second linear layer as:

$$\text{Cov}(h_{\text{MLP}}) = \text{Cov}(vW_2) = W_2^\top \text{Cov}(v) W_2.$$

Under the assumption that the activation function $\phi$ essentially acts as a scaling factor $\tau$ on the covariance matrix (which holds exactly for linear activations or as a linear approximation for nonlinear ones), we substitute $\text{Cov}(v) \approx \tau \text{Cov}(u)$:

$$\text{Cov}(h_{\text{MLP}}) \approx W_2^\top (\tau \text{Cov}(u)) W_2 = \tau W_2^\top \text{Cov}(zW_1) W_2.$$

Further expanding $\text{Cov}(u) = \text{Cov}(zW_1) = W_1^\top \text{Cov}(z)W_1$, we obtain:

$$\text{Cov}(h_{\text{MLP}}) \approx \tau W_2^\top (W_1^\top \text{Cov}(z)W_1)W_2 = \tau(W_1 W_2)^\top \text{Cov}(z)(W_1 W_2).$$

Assuming the input $z$ consists of uncorrelated elements with uniform variance $s_z^2$ (consistent with the output of BHyT where $s_z^2 \approx \lambda^2/\kappa^2$), the covariance of the input is $\text{Cov}(z) = s_z^2 I_d$. Substituting this into the expression gives:

$$\text{Cov}(h_{\text{MLP}}) \approx \tau s_z^2 (W_1 W_2)^\top (W_1 W_2).$$

Finally, taking the trace to find the average sample variance:

$$\tilde{s}_{h_{\text{MLP}}}^2 = \frac{1}{d}\text{Tr}\left(\tau s_z^2 (W_1 W_2)^\top (W_1 W_2)\right) = \tau \frac{s_z^2}{d}\text{Tr}((W_1 W_2)^\top (W_1 W_2)).$$

By the definition of the Frobenius norm, $\text{Tr}(A^\top A) = \|A\|_F^2$, which leads to the final approximation:

$$\tilde{s}_{h_{\text{MLP}}}^2 \approx \tau \frac{s_z^2}{d}\|W_1 W_2\|_F^2.$$

$\square$

While Llama architectures employ SwiGLU (Shazeer, 2020) variants involving three matrices (gate, up, down), for theoretical tractability, we analyze the standard MLP formulation parameterized by $W_1$ and $W_2$. We assume this variance propagation logic generalizes to gated units by treating the gating mechanism as part of the effective activation scaling factor $\tau$.

### B.5. Proof of Theorem 3.5

#### B.5.1. VARIANCE BOUND FOR BHYT

**Lemma B.3** (Variance bound for $\tanh$ in BHyT). *Let $x$ be a scalar random variable with a probability density function symmetric about zero, such that $\mathbb{E}[x] = 0$ and $\text{Var}(x) = s_x^2$. Define the scaling factor $\alpha = \frac{\lambda}{\kappa s_x}$ for hyperparameters $\lambda, \kappa > 0$. Assuming the bound $|\alpha x| \leq \lambda$ holds (or conditioning on this event), the variance of the activation satisfies:*

$$\left(\frac{\tanh(\lambda)}{\lambda}\right)^2 \frac{\lambda^2}{\kappa^2} \leq \text{Var}(\tanh(\alpha x)) \leq \frac{\lambda^2}{\kappa^2}.$$

*Proof.* First, we establish point-wise inequalities for the function $h(t) = \tanh(t)$ on the interval $[-\lambda, \lambda]$.

Upper Bound: For any real number $t$, it is a standard property that $|\tanh(t)| \leq |t|$. Squaring both sides gives:

$$\tanh^2(t) \leq t^2.$$

Substituting $t = \alpha x$, we obtain $\tanh^2(\alpha x) \leq (\alpha x)^2$. Taking the expectation on both sides:

$$\mathbb{E}[\tanh^2(\alpha x)] \leq \mathbb{E}[(\alpha x)^2] = \alpha^2 \mathbb{E}[x^2] = \alpha^2 s_x^2.$$

Substituting $\alpha = \frac{\lambda}{\kappa s_x}$, we get $\alpha^2 s_x^2 = \left(\frac{\lambda}{\kappa s_x}\right)^2 s_x^2 = \frac{\lambda^2}{\kappa^2}$. Since $\mathbb{E}[x] = 0$ and the distribution is symmetric, $\mathbb{E}[\tanh(\alpha x)] = 0$, implying $\text{Var}(\tanh(\alpha x)) = \mathbb{E}[\tanh^2(\alpha x)]$. Thus the upper bound of $\text{Var}(\tanh(\alpha x))$ is

$$\text{Var}(\tanh(\alpha x)) \leq \frac{\lambda^2}{\kappa^2}.$$

Lower Bound: Consider $t \in [0, \lambda]$. Since $h(t) = \tanh(t)$ is concave on $[0, \infty)$ and $h(0) = 0$, the secant line connecting $(0, 0)$ and $(\lambda, \tanh(\lambda))$ lies below the graph of $h(t)$ for $t \in [0, \lambda]$. The equation of this line is $y = \frac{\tanh(\lambda)}{\lambda}t$. Therefore:

$$\tanh(t) \geq \frac{\tanh(\lambda)}{\lambda}t \quad \text{for } t \in [0, \lambda].$$

By the odd symmetry of $\tanh(t)$, for $t \in [-\lambda, 0]$, we have $\tanh(t) \le \frac{\tanh(\lambda)}{\lambda} t$ (since both sides are negative). Squaring both cases yields the same inequality for all $t \in [-\lambda, \lambda]$:

$$\tanh^2(t) \ge \left(\frac{\tanh(\lambda)}{\lambda}\right)^2 t^2.$$

Substituting $t = \alpha x$ and taking the expectation (under the assumption $|\alpha x| \le \lambda$):

$$\mathbb{E}[\tanh^2(\alpha x)] \ge \left(\frac{\tanh(\lambda)}{\lambda}\right)^2 \mathbb{E}[(\alpha x)^2] = \left(\frac{\tanh(\lambda)}{\lambda}\right)^2 \frac{\lambda^2}{\kappa^2}.$$

$\square$

### B.5.2. VARIANCE RECURSION WITH RESIDUAL CONNECTIONS

We next derive the variance recursion for a generic residual block of the form

$$x'_\ell = x_\ell + h_\ell, \qquad x_{\ell+1} = x'_\ell + g_\ell,$$

where $h_\ell = \text{Attn}(f(x_\ell))$ and $g_\ell = \text{MLP}(f(x'_\ell))$. Here, $f(\cdot)$ denotes a normalization layer (e.g., RMSNorm, BHyT, etc.).

**Lemma B.4** (Generic variance recursion)**.** *Let $\tilde{s}^2_{x_\ell}$ and $\tilde{s}^2_{x'_\ell}$ denote the average sample variances of the residual stream before and after the attention sublayer at layer $\ell$. Based on the approximations in Theorems 3.4 and B.2, we define the structural amplification factors as:*

$$C_{\text{Attn}} := \frac{1}{Td}\|W_V W_O\|_F^2, \quad C_{\text{MLP}} := \frac{\tau}{d}\|W_1 W_2\|_F^2.$$

*Let $\tilde{s}^2_{f(x_\ell)}$ and $\tilde{s}^2_{f(x'_\ell)}$ be the variances of the normalization layer outputs. Assuming linear correlations parameterized by coefficients $\rho_1$ and $\rho_2$, the variances satisfy the recursion:*

$$\tilde{s}^2_{x'_\ell} = \tilde{s}^2_{x_\ell} + C_{\text{Attn}}\tilde{s}^2_{f(x_\ell)} + 2\rho_1 \tilde{s}_{x_\ell}\sqrt{C_{\text{Attn}}}\tilde{s}_{f(x_\ell)}, \tag{8}$$

$$\tilde{s}^2_{x_{\ell+1}} = \tilde{s}^2_{x'_\ell} + C_{\text{MLP}}\tilde{s}^2_{f(x'_\ell)} + 2\rho_2 \tilde{s}_{x'_\ell}\sqrt{C_{\text{MLP}}}\tilde{s}_{f(x'_\ell)}. \tag{9}$$

*Proof.* We analyze the variance accumulation through the residual connections. For the attention sublayer, the residual update is $x'_\ell = x_\ell + h_\ell$. By the properties of variance, we have:

$$\text{Var}(x'_\ell) = \text{Var}(x_\ell) + \text{Var}(h_\ell) + 2\text{Cov}(x_\ell, h_\ell).$$

Using the average sample variance notation $\tilde{s}^2$, and applying the result from Theorem 3.4, the variance of the attention output is approximated by the product of the structural factor and the input variance: $\tilde{s}^2_{h_\ell} \approx C_{\text{Attn}}\tilde{s}^2_{f(x_\ell)}$. For the covariance term, we introduce the correlation coefficient $\rho_1$ between the residual stream $x_\ell$ and the attention sublayer output $h_\ell$, such that:

$$\text{Cov}(x_\ell, h_\ell) = \rho_1 \sqrt{\text{Var}(x_\ell)}\sqrt{\text{Var}(h_\ell)} = \rho_1 \tilde{s}_{x_\ell}\tilde{s}_{h_\ell} \approx \rho_1 \tilde{s}_{x_\ell}\sqrt{C_{\text{Attn}}}\tilde{s}_{f(x_\ell)}.$$

Substituting these terms back into the variance equation yields the first recursion formula. The derivation for the MLP sublayer follows strictly analogous steps. Starting from $x_{\ell+1} = x'_\ell + g_\ell$, we expand the variance as $\tilde{s}^2_{x_{\ell+1}} = \tilde{s}^2_{x'_\ell} + \tilde{s}^2_{g_\ell} + 2\text{Cov}(x'_\ell, g_\ell)$. Invoking Theorem B.2, the MLP output variance is $\tilde{s}^2_{g_\ell} \approx C_{\text{MLP}}\tilde{s}^2_{f(x'_\ell)}$. Defining $\rho_2$ as the correlation coefficient between $x'_\ell$ and $g_\ell$, the covariance is modeled as $\rho_2 \tilde{s}_{x'_\ell}\sqrt{C_{\text{MLP}}}\tilde{s}_{f(x'_\ell)}$. $\square$

### B.5.3. SPECIALIZATION TO RMSNORM, LNS AND BHyT

In this section, we instantiate the generic variance recursion from the assumptions and results of Theorems 3.4 and B.2, for three concrete choices of $f(\cdot)$: RMSNorm, LNS (Sun et al., 2026), and BHyT. We follow the derivation in Appendix B.5.2.

**Notation.** We denote by

$$s^2_{x_\ell} = \text{Var}(x_\ell), \quad s^2_{x'_\ell} = \text{Var}(x'_\ell), \quad s^2_{f(x_\ell)} = \text{Var}(f(x_\ell)), \quad s^2_{f(x'_\ell)} = \text{Var}(f(x'_\ell)).$$

The output variances of the attention and MLP sublayers are $s^2_{\text{Attn},\ell} = \text{Var}(\text{Attn}(f(x_\ell)))$ and $s^2_{\text{MLP},\ell} = \text{Var}(\text{MLP}(f(x'_\ell)))$, respectively.

B.5.4. VARIANCE RECURSION AND PRODUCT FORM

To simplify the recursion, we define the normalized variance gains $\pi_{\text{Attn},\ell}$ and $\pi_{\text{MLP},\ell}$, which represent the relative variance contribution of each sublayer scaled by the structural constants $C_{\text{Attn}}$ and $C_{\text{MLP}}$:

$$\pi_{\text{Attn},\ell} := \frac{C_{\text{Attn}}\tilde{s}^2_{f(x_\ell)}}{\tilde{s}^2_{x_\ell}}, \quad \pi_{\text{MLP},\ell} := \frac{C_{\text{MLP}}\tilde{s}^2_{f(x'_\ell)}}{\tilde{s}^2_{x'_\ell}}.$$

We also define the scalar amplification function $\delta(\rho, u)$ as:

$$\delta(\rho, \pi) := 1 + \pi + 2\rho\sqrt{\pi}.$$

Using Lemma B.4, the layer-wise variance amplification ratios can be written compactly as:

$$\frac{\tilde{s}^2_{x'_\ell}}{\tilde{s}^2_{x_\ell}} = 1 + \frac{C_{\text{Attn}}\tilde{s}^2_{f(x_\ell)}}{\tilde{s}^2_{x_\ell}} + 2\rho_1\sqrt{\frac{C_{\text{Attn}}\tilde{s}^2_{f(x_\ell)}}{\tilde{s}^2_{x_\ell}}} = \delta(\rho_1, \pi_{\text{Attn},\ell}), \tag{10}$$

$$\frac{\tilde{s}^2_{x_{\ell+1}}}{\tilde{s}^2_{x'_\ell}} = 1 + \frac{C_{\text{MLP}}\tilde{s}^2_{f(x'_\ell)}}{\tilde{s}^2_{x'_\ell}} + 2\rho_2\sqrt{\frac{C_{\text{MLP}}\tilde{s}^2_{f(x'_\ell)}}{\tilde{s}^2_{x'_\ell}}} = \delta(\rho_2, \pi_{\text{MLP},\ell}). \tag{11}$$

Combining the amplification factors for both sublayers, the total variance growth for a single block at layer $\ell$ is:

$$\frac{\tilde{s}^2_{x_{\ell+1}}}{\tilde{s}^2_{x_\ell}} = \delta(\rho_1, \pi_{\text{Attn},\ell}) \cdot \delta(\rho_2, \pi_{\text{MLP},\ell}).$$

Iterating this relation from layer $\ell = 1$ to $L - 1$, we obtain the closed-form expression for the variance at the final layer $L$:

$$\tilde{s}^2_{x_L} = \tilde{s}^2_{x_1} \prod_{\ell=1}^{L-1} \delta(\rho_1, \pi_{\text{Attn},\ell}) \cdot \delta(\rho_2, \pi_{\text{MLP},\ell}).$$

This product form allows us to analyze the signal propagation behavior by simply substituting the specific normalized gains $\pi$ corresponding to each normalization method (RMSNorm, LNS, and BHyT).

**Lemma B.5** (Depth-wise variance under RMSNorm). *Let $f(x)$ be RMSNorm applied coordinate-wise to a $d$-dimensional vector $x$, defined by learnable scales $\gamma \in \mathbb{R}^d$. Let $\tilde{s}^2_{x_\ell}$ and $\tilde{s}^2_{x'_\ell}$ denote the average sample variances of the residual stream before and after the attention sublayer at layer $\ell$. We define the average squared scale as $\overline{\gamma^2} = \frac{1}{d}\sum_{i=1}^d \gamma_i^2$. Using the structural constants $C_{\text{Attn}}$ and $C_{\text{MLP}}$ defined in Lemma B.4, we define the normalized variance gains for RMSNorm as:*

$$\pi_{\text{Attn},\ell}^{RMS} := \frac{C_{\text{Attn}}\overline{\gamma^2_{\text{Attn}}}}{\tilde{s}^2_{x_\ell}}, \quad \pi_{\text{MLP},\ell}^{RMS} := \frac{C_{\text{MLP}}\overline{\gamma^2_{\text{MLP}}}}{\tilde{s}^2_{x'_\ell}}.$$

*Let $\delta(\rho, \pi) = 1 + \pi + 2\rho\sqrt{\pi}$ be the scalar amplification function. Then, the depth-wise variance accumulation under RMSNorm satisfies:*

$$\tilde{s}^2_{x_L} = \tilde{s}^2_{x_1} \prod_{\ell=1}^{L-1} \delta(\rho_1, \pi_{\text{Attn},\ell}^{RMS}) \delta(\rho_2, \pi_{\text{MLP},\ell}^{RMS}).$$

*Proof.* We first determine the variance of the RMSNorm output. For a vector $x$ with zero-mean, RMSNorm is defined as $\text{RMSNorm}(x) = \gamma \odot \frac{x}{\sqrt{\frac{1}{d}\sum_{j=1}^d x_j^2}}$. Squaring and taking the expectation, and utilizing the symmetry of the isotropic input distribution (where $\mathbb{E}[\frac{x_i^2}{\frac{1}{d}\sum x_j^2}] = 1$), we obtain:

$$\mathbb{E}[\text{RMSNorm}(x)_i^2] = \gamma_i^2 \cdot \mathbb{E}\left[\frac{x_i^2}{\frac{1}{d}\sum_{j=1}^d x_j^2}\right] = \gamma_i^2.$$

Thus, the average sample variance of the output is independent of the input variance magnitude and is determined solely by the scale parameters:

$$\tilde{s}^2_{f(x)} = \frac{1}{d} \sum_{i=1}^{d} \gamma_i^2 = \overline{\gamma^2}.$$

Consequently, at any layer $\ell$, the variances of the normalization outputs feeding into the Attention and MLP blocks are identical: $\tilde{s}^2_{f(x_\ell)} = \overline{\gamma^2_{\text{Attn}}}$ and $\tilde{s}^2_{f(x'_\ell)} = \overline{\gamma^2_{\text{MLP}}}$. Substituting these fixed variance terms into the definition of the normalized variance gains (from Lemma B.4) yields the specific forms for RMSNorm:

$$\pi^{\text{RMS}}_{\text{Attn},\ell} = \frac{C_{\text{Attn}}\tilde{s}^2_{f(x_\ell)}}{\tilde{s}^2_{x_\ell}} = \frac{C_{\text{Attn}}\overline{\gamma^2_{\text{Attn}}}}{\tilde{s}^2_{x_\ell}}, \quad \pi^{\text{RMS}}_{\text{MLP},\ell} = \frac{C_{\text{MLP}}\tilde{s}^2_{f(x'_\ell)}}{\tilde{s}^2_{x'_\ell}} = \frac{C_{\text{MLP}}\overline{\gamma^2_{\text{MLP}}}}{\tilde{s}^2_{x'_\ell}}.$$

$\square$

**Lemma B.6** (Depth-wise variance under LNS). *Let $f(x)$ be the LNS (Sun et al., 2026) operation, defined as RMSNorm followed by a depth-dependent scalar $1/\sqrt{\ell}$ at layer $\ell$. Using the same notation as in Lemma B.5, the normalized variance gains for LNS are given by:*

$$\pi^{LNS}_{Attn,\ell} \approx \frac{C_{Attn}\overline{\gamma^2_{Attn}}}{\ell \cdot \tilde{s}^2_{x_\ell}}, \quad \pi^{LNS}_{MLP,\ell} \approx \frac{C_{MLP}\overline{\gamma^2_{MLP}}}{\ell \cdot \tilde{s}^2_{x'_\ell}}.$$

*Consequently, the depth-wise variance accumulation under LNS satisfies:*

$$\tilde{s}^2_{x_L} = \tilde{s}^2_{x_1} \prod_{\ell=1}^{L-1} \delta(\rho_1, \pi^{LNS}_{Attn,\ell})\delta(\rho_2, \pi^{LNS}_{MLP,\ell}).$$

*Proof.* By Lemma B.5, the standard RMSNorm operation produces an output with average sample variance $\tilde{s}^2_{\text{RMS}(x)} = \overline{\gamma^2}$, which is independent of the input variance. LNS introduces an additional scalar multiplication by $1/\sqrt{\ell}$ after RMSNorm. Since variance scales quadratically with the multiplier, the variance of the LNS output at layer $\ell$ becomes:

$$\tilde{s}^2_{f(x_\ell)} = \text{Var}\left(\frac{1}{\sqrt{\ell}}\text{RMSNorm}(x_\ell)\right) = \frac{1}{\ell}\tilde{s}^2_{\text{RMS}(x_\ell)} = \frac{\overline{\gamma^2}}{\ell}.$$

Substituting this result into the definition of the normalized variance gain $\pi_{\text{Attn},\ell} = C_{\text{Attn}}\tilde{s}^2_{f(x_\ell)}/\tilde{s}^2_{x_\ell}$, we obtain:

$$\pi^{\text{LNS}}_{\text{Attn},\ell} = \frac{C_{\text{Attn}}(\overline{\gamma^2}/\ell)}{\tilde{s}^2_{x_\ell}} = \frac{C_{\text{Attn}}\overline{\gamma^2}}{\ell \cdot \tilde{s}^2_{x_\ell}}.$$

The derivation for the MLP sublayer term $\pi^{\text{LNS}}_{\text{MLP},\ell}$ follows identically. $\square$

**Lemma B.7** (Depth-wise variance bounds under BHyT). *Let $f(x)$ be the BHyT, parameterized by hyperparameters $(\lambda, \kappa)$ and a learnable scale vector $\gamma$. Let $\overline{\gamma^2}$ be the average squared value of $\gamma$. Based on the variance bounds derived in Lemma B.3, we define the lower and upper bound approximations for the normalized variance gains at layer $\ell$ as:*

$$\pi^{BHyT,low}_{Attn,\ell} := \frac{C_{Attn}\overline{\gamma^2}}{\tilde{s}^2_{x_\ell}} \cdot \left(\frac{\tanh(\lambda_{Attn})}{\lambda_{Attn}}\right)^2 \cdot \frac{\lambda^2_{Attn}}{\kappa^2}, \quad \pi^{BHyT,up}_{Attn,\ell} := \frac{C_{Attn}\overline{\gamma^2}}{\tilde{s}^2_{x_\ell}} \cdot \frac{\lambda^2_{Attn}}{\kappa^2}.$$

*Analogous definitions apply for $\pi^{BHyT,low}_{MLP,\ell}$ and $\pi^{BHyT,up}_{MLP,\ell}$ by substituting the appropriate structural constant $C_{MLP}$ and input variance $\tilde{s}^2_{x'_\ell}$. Using the scalar amplification function $\delta(\rho, \pi) = 1 + \pi + 2\rho\sqrt{\pi}$, the depth-wise variance accumulation under BHyT satisfies the following two-sided bounds:*

$$\tilde{s}^2_{x_1} \prod_{\ell=1}^{L-1} \delta(\rho_1, \pi^{BHyT,low}_{Attn,\ell})\delta(\rho_2, \pi^{BHyT,low}_{MLP,\ell}) \leq \tilde{s}^2_{x_L} \leq \tilde{s}^2_{x_1} \prod_{\ell=1}^{L-1} \delta(\rho_1, \pi^{BHyT,up}_{Attn,\ell})\delta(\rho_2, \pi^{BHyT,up}_{MLP,\ell}).$$

*Proof.* We start by examining the variance of the BHyT output. BHyT is defined as $f(x) = \gamma \odot \tanh(\frac{\lambda}{\kappa s_x} x)$. According to Lemma B.3, the variance of the tanh component, denoted as $z = \tanh(\frac{\lambda}{\kappa s_x} x)$, is bounded by:

$$\left(\frac{\tanh(\lambda)}{\lambda}\right)^2 \frac{\lambda^2}{\kappa^2} \leq \mathrm{Var}(z) \leq \frac{\lambda^2}{\kappa^2}.$$

Since the learnable scale $\gamma$ acts element-wise, the average sample variance of the BHyT output $\tilde{s}^2_{f(x)}$ is the product of the average squared scale $\overline{\gamma^2}$ and the variance of the tanh activation. Therefore, the output variance at layer $\ell$ is bounded by:

$$\overline{\gamma^2}\left(\frac{\tanh(\lambda)}{\lambda}\right)^2 \frac{\lambda^2}{\kappa^2} \leq \tilde{s}^2_{f(x_\ell)} \leq \overline{\gamma^2}\frac{\lambda^2}{\kappa^2}.$$

Substituting these inequalities into the definition of the normalized variance gain $\pi_{\mathrm{Attn},\ell} = C_{\mathrm{Attn}}\tilde{s}^2_{f(x_\ell)}/\tilde{s}^2_{x_\ell}$ yields the definitions for $\pi^{\mathrm{BHyT,low}}_{\mathrm{Attn},\ell}$ and $\pi^{\mathrm{BHyT,up}}_{\mathrm{Attn},\ell}$ stated in Lemma B.7. Crucially, the scalar amplification function $\delta(\rho, \pi) = 1 + \pi + 2\rho\sqrt{\pi}$ is monotonically increasing with respect to $\pi$ for $\pi > 0$ and $\rho \geq 0$. This monotonicity allows us to propagate the inequalities through the product-recursive form derived in Appendix B.5.4. Specifically:

$$\delta(\rho_1, \pi^{\mathrm{BHyT,low}}_{\mathrm{Attn},\ell}) \leq \delta(\rho_1, \pi_{\mathrm{Attn},\ell}) \leq \delta(\rho_1, \pi^{\mathrm{BHyT,up}}_{\mathrm{Attn},\ell}).$$

Applying this relationship to both the Attention and MLP sublayers across all layers $\ell = 1, \ldots, L-1$ proves the stated lower and upper bounds for the final depth-wise variance $\tilde{s}^2_{x_L}$. $\qquad\square$

### B.5.5. PROOF OF THEOREM 3.5

**Theorem 3.5** (Finite-depth variance bound of BHyT). *For a network of depth L, if the BHyT hyperparameters satisfy $\lambda/\kappa < 1/\sqrt{L}$, then the output variance under BHyT is strictly smaller than that under LNS (Sun et al., 2026) for every layer $\ell$ up to depth L, i.e., for all $1 \leq \ell \leq L$:*

$$\tilde{s}^{BHyT}_{x_\ell} < \tilde{s}^{LNS}_{x_\ell}, \quad \forall \ell \in \{1, \ldots, L\}. \tag{7}$$

*Proof.* To compare the output variances of BHyT and LNS, we examine their respective layer-wise variance amplification factors derived in Lemma B.6 and Lemma B.7. Recall the recursive product form:

$$\tilde{s}^2_{x_L} = \tilde{s}^2_{x_1} \prod_{\ell=1}^{L-1} \delta(\rho_1, \pi_{\mathrm{Attn},\ell})\delta(\rho_2, \pi_{\mathrm{MLP},\ell}).$$

Since the scalar amplification function $\delta(\rho, \pi) = 1 + \pi + 2\rho\sqrt{\pi}$ is monotonically increasing with respect to $\pi$ for $\pi > 0$ and $\rho \geq 0$, it suffices to compare the normalized variance gain terms $\pi$ at each layer $\ell$. From Lemma B.6, the variance gain for LNS at layer $\ell$ scales as:

$$\pi^{\mathrm{LNS}}_{\mathrm{Attn},\ell} = \frac{C_{\mathrm{Attn}}\overline{\gamma^2_{\mathrm{Attn}}}}{\ell \cdot \tilde{s}^2_{x_\ell}}, \quad \pi^{\mathrm{LNS}}_{\mathrm{MLP},\ell} = \frac{C_{\mathrm{MLP}}\overline{\gamma^2_{\mathrm{MLP}}}}{\ell \cdot \tilde{s}^2_{x'_\ell}}.$$

From Lemma B.7, the upper bound of the variance gain for BHyT at layer $\ell$ is determined by:

$$\pi^{\mathrm{BHyT,up}}_{\mathrm{Attn},\ell} = \frac{C_{\mathrm{Attn}}\overline{\gamma^2_{\mathrm{Attn}}}(\frac{\lambda^2_{\mathrm{Attn}}}{\kappa^2})}{\tilde{s}^2_{x_\ell}}, \quad \pi^{\mathrm{BHyT,up}}_{\mathrm{MLP},\ell} = \frac{C_{\mathrm{MLP}}\overline{\gamma^2_{\mathrm{MLP}}}(\frac{\lambda^2_{\mathrm{MLP}}}{\kappa^2})}{\tilde{s}^2_{x'_\ell}}.$$

For the output variance of BHyT to be strictly smaller than that of LNS, the amplification factor of BHyT must be smaller than that of LNS at each layer. Comparing the Attention terms (the MLP terms follow analogously), we require:

$$\delta(\rho_1, \pi^{\mathrm{BHyT,up}}_{\mathrm{Attn},\ell}) < \delta(\rho_1, \pi^{\mathrm{LNS}}_{\mathrm{Attn},\ell}).$$

Due to the monotonicity of $\delta$, this inequality holds if and only if $\pi^{\mathrm{BHyT,up}}_{\mathrm{Attn},\ell} < \pi^{\mathrm{LNS}}_{\mathrm{Attn},\ell}$. Substituting the explicit forms into the comparison:

$$\frac{C_{\mathrm{Attn}}\overline{\gamma^2_{\mathrm{Attn}}}\frac{\lambda^2_{\mathrm{Attn}}}{\kappa^2}}{\tilde{s}^2_{x_\ell}} < \frac{C_{\mathrm{Attn}}\overline{\gamma^2_{\mathrm{Attn}}}}{\ell \cdot \tilde{s}^2_{x_\ell}}.$$

Canceling the common positive terms ($C_{\text{Attn}}, \overline{\gamma_{\text{Attn}}^2}, \tilde{s}_{x_\ell}^2$), we obtain the condition for the scaling factors:

$$\frac{\lambda_{\text{Attn}}^2}{\kappa^2} < \frac{1}{\ell}.$$

Taking the square root implies $\frac{\lambda_{\text{Attn}}}{\kappa} < \frac{1}{\sqrt{\ell}}$. The term $\frac{1}{\sqrt{\ell}}$ is a decreasing function of $\ell$. Therefore, satisfying this condition for the maximum depth $L$ (i.e., $\frac{\lambda_{\text{Attn}}}{\kappa} < \frac{1}{\sqrt{L}}$) ensures that it holds for all shallower layers $\ell < L$. Consequently, if this condition is met, the per-layer variance multiplier of BHyT is strictly smaller than that of LNS for all layers, leading to the final inequality:

$$\tilde{s}_{x_L}^{\text{BHyT}} < \tilde{s}_{x_L}^{\text{LNS}}.$$

$\square$

## C. Hyperparameter Search and Experimental Setup

**Hyperparameter Search Protocol**    We perform hyperparameter tuning within a shared sweep range to ensure a fair comparison across all normalization methods. For each method, we identify the optimal configuration by selecting the hyperparameters that achieve the lowest evaluation loss after the Llama-1B model has completed 20K pretraining steps. This selected configuration is then used for the full pretraining run and subsequently for supervised fine-tuning (SFT).

### C.1. Hyperparameters for Pretraining

**Common sweep ranges**

- **Learning rate (LR):** $\{\texttt{1e-4, 3e-4, 5e-4, 1e-3, 3e-3}\}$
- **Weight decay (WD):** $\{\texttt{0.0, 0.1}\}$
- **Min LR ratio:** $\{\texttt{1e-1, 1e-2}\}$
- **Warmup ratio:** $\{\texttt{5e-2, 1e-1}\}$

**Method-specific sweep ranges**

- **BHyT:** initial values of $\lambda$: $\{\texttt{1.0, 2.0, 3.0, 4.0, 5.0}\}$
- **DyT:** learnable tanh-scaling parameter initialized using recommended values from the DyT (Zhu et al., 2025): for Llama-374M and Llama-1B, we set the scalars to $\texttt{1.0}$ (before Attention), $\texttt{0.5}$ (before MLP), and $\texttt{0.5}$ (last layer), while for Llama-3B we use $\texttt{0.2}, \texttt{0.05}$, and $\texttt{0.05}$, respectively.

### C.2. SFT Hyperparameter Sweep

- **Learning rate (LR):** $\{\texttt{1e-7, 5e-7, 1e-6, 1e-5, 5e-5, 1e-4}\}$
- **Weight decay (WD):** $\{\texttt{0.0, 0.1}\}$
- **Min LR ratio:** $\{\texttt{1e-1, 1e-2, 1e-3}\}$
- **Warmup ratio:** $\{\texttt{3e-2, 5e-2, 1e-1}\}$

### C.3. Final Hyperparameter Configurations

- RMSNorm: {LR: $\texttt{3e-4}$, WD: $\texttt{0.1}$, Min LR ratio: $\texttt{1e-1}$, Warmup ratio: $\texttt{1e-1}$}
- Peri-LN: {LR: $\texttt{3e-4}$, WD: $\texttt{0.1}$, Min LR ratio: $\texttt{1e-1}$, Warmup ratio: $\texttt{1e-1}$}
- LNS: {LR: $\texttt{5e-4}$, WD: $\texttt{0.1}$, Min LR ratio: $\texttt{1e-1}$, Warmup ratio: $\texttt{5e-2}$}
- DyT: {LR: $\texttt{1e-4}$, WD: $\texttt{0.1}$, Min LR ratio: $\texttt{1e-1}$, Warmup ratio: $\texttt{1e-1}$}
- BHyT: {LR: $\texttt{5e-4}$, WD: $\texttt{0.1}$, Min LR ratio: $\texttt{1e-2}$, Warmup ratio: $\texttt{1e-1}$, $\lambda$ values: BHyT$_{\text{Attn}}$: $\texttt{2.0}$, BHyT$_{\text{MLP}}$: $\texttt{1.0}$}

## C.4. Hardware and system configuration

- **Pretraining and SFT framework:** Llama-Factory (Zheng et al., 2024)

- **Llama-1B GPU:** RTX A6000

- **Llama-3B GPU:** RTX PRO 6000 Blackwell

- **Parallel computation:** For efficient parallel computation, we use FlashAttention-2 (Dao, 2023)

- **Distributed training:** DeepSpeed ZeRO-2

## C.5. Full-baseline comparison on Llama-3B under a short-budget setting

*Table 6.* Pretraining (PT)-only evaluation for Llama-3B; downstream benchmarks in a 5-shot setting, averaged over five training seeds. The results demonstrate that BHyT achieves performance comparable to or superior to strong baselines while maintaining high computational speed. PPL denotes the perplexity score.

| | | | | | | | Llama-3B | | | | |
|---|---|---|---|---|---|---|---|---|---|---|---|
| Method | PT Train Loss | PT Eval Loss | PT Eval PPL | Arc-e | PIQA | Hellaswag | OpenBookQA | Winogrande | MMLU | BoolQ | Avg. |
| RMSNorm | 3.203 | 3.180 | 24.040 | $31.01_{\pm0.21}$ | $\mathbf{66.57}_{\pm0.41}$ | $\mathbf{41.52}_{\pm0.25}$ | $\mathbf{33.92}_{\pm0.33}$ | $50.50_{\pm0.78}$ | $25.67_{\pm0.02}$ | $37.94_{\pm0.12}$ | $40.89_{\pm0.18}$ |
| Peri-LN | 3.165 | 3.142 | 23.156 | $31.82_{\pm0.38}$ | $64.52_{\pm0.24}$ | $36.05_{\pm0.11}$ | $32.28_{\pm0.59}$ | $49.30_{\pm0.52}$ | $25.99_{\pm0.00}$ | $57.46_{\pm0.31}$ | $42.49_{\pm0.25}$ |
| LNS | 3.160 | 3.139 | 23.091 | $\mathbf{31.88}_{\pm0.42}$ | $64.68_{\pm0.46}$ | $36.25_{\pm0.10}$ | $32.45_{\pm0.53}$ | $51.18_{\pm0.75}$ | $\mathbf{26.87}_{\pm0.01}$ | $37.83_{\pm0.08}$ | $41.16_{\pm0.11}$ |
| DyT | 3.877 | 3.855 | 47.244 | $27.69_{\pm0.27}$ | $59.20_{\pm0.15}$ | $25.96_{\pm0.17}$ | $31.85_{\pm0.38}$ | $49.17_{\pm1.28}$ | $25.87_{\pm0.02}$ | $48.05_{\pm0.56}$ | $38.26_{\pm0.12}$ |
| BHyT | $\mathbf{3.133}$ | $\mathbf{3.107}$ | $\mathbf{22.346}$ | $31.84_{\pm0.15}$ | $65.08_{\pm0.39}$ | $36.48_{\pm0.04}$ | $31.76_{\pm0.33}$ | $\mathbf{51.62}_{\pm0.66}$ | $25.70_{\pm0.01}$ | $\mathbf{60.84}_{\pm0.29}$ | $\mathbf{43.33}_{\pm0.16}$ |

Table 6 shows the pretraining evaluation results, including training loss. The results confirm that BHyT consistently achieves the lowest pretraining loss and superior average accuracy, validating its robust scalability and performance advantage.

*Table 7.* Supervised fine-tuning (SFT) results across five training seeds for Llama-3B. Reported values include SFT losses and 5-shot downstream benchmark accuracies; BHyT attains lower SFT loss and competitive performance, indicating effective transfer of its pretraining stability to instruction-tuned settings.

| | | | | | | | Llama-3B | | | | |
|---|---|---|---|---|---|---|---|---|---|---|---|
| Method | PT Eval Loss | SFT Train Loss | SFT Eval Loss | Arc-e | PIQA | Hellaswag | OpenBookQA | Winogrande | MMLU | BoolQ | Avg. |
| RMSNorm | 3.272 | $2.646_{\pm0.011}$ | $3.217_{\pm0.130}$ | $\mathbf{37.67}_{\pm0.23}$ | $66.96_{\pm0.29}$ | $31.70_{\pm0.12}$ | $31.12_{\pm0.18}$ | $\mathbf{51.22}_{\pm1.60}$ | $22.95_{\pm0.04}$ | $53.31_{\pm2.02}$ | $39.83_{\pm0.21}$ |
| Peri-LN | 3.279 | $\mathbf{2.614}_{\pm0.011}$ | $3.178_{\pm0.132}$ | $36.80_{\pm0.27}$ | $\mathbf{67.05}_{\pm0.12}$ | $37.13_{\pm0.09}$ | $\mathbf{31.44}_{\pm0.61}$ | $49.06_{\pm0.40}$ | $22.94_{\pm0.02}$ | $52.75_{\pm1.54}$ | $42.45_{\pm0.28}$ |
| LNS | $3.271$ | $2.652_{\pm0.011}$ | $3.157_{\pm0.132}$ | $34.49_{\pm0.17}$ | $65.59_{\pm0.16}$ | $37.12_{\pm0.09}$ | $28.96_{\pm0.54}$ | $50.61_{\pm0.55}$ | $22.99_{\pm0.02}$ | $52.69_{\pm0.66}$ | $41.78_{\pm0.20}$ |
| DyT | 3.855 | $3.361_{\pm0.010}$ | $3.971_{\pm0.132}$ | $29.78_{\pm0.40}$ | $58.42_{\pm0.53}$ | $25.95_{\pm0.17}$ | $28.28_{\pm0.90}$ | $49.53_{\pm0.79}$ | $\mathbf{23.46}_{\pm0.13}$ | $56.64_{\pm1.02}$ | $38.87_{\pm0.17}$ |
| BHyT | $\mathbf{3.254}$ | $2.693_{\pm0.012}$ | $\mathbf{3.130}_{\pm0.133}$ | $34.61_{\pm0.20}$ | $66.47_{\pm0.27}$ | $36.95_{\pm0.12}$ | $29.68_{\pm0.46}$ | $51.14_{\pm1.02}$ | $23.07_{\pm0.05}$ | $\mathbf{58.21}_{\pm0.81}$ | $\mathbf{42.88}_{\pm0.19}$ |

We apply LoRA-based supervised fine-tuning to assess whether the stability achieved during pretraining carries over to instruction-tuned performance. Table 7 summarizes the SFT losses and downstream benchmark accuracies for Llama-3B. Across all methods, BHyT maintains the stable optimization observed during pretraining and effectively transfers this advantage to the SFT stage. Notably, BHyT attains lower SFT loss and competitive accuracy on a broad set of benchmarks, demonstrating that its depth-wise stability and controlled activation dynamics continue to support reliable adaptation beyond pretraining.

## C.6. Short-budget setting for the variance approximation ablation

The variance approximation ablation in Table 4 uses a short-budget pretraining setting to isolate the computational effect of the approximation at lower experimental cost. This setting is separate from the main 20B-token experiments in Section 4, which provide the primary PT-only and post-SFT comparisons.

We pretrain Llama-1B models on C4 with sequence length 1024, batch size 32, and 50,000 optimization steps, corresponding to approximately 1.64B training tokens. We compare exact-statistic variants with their approximate counterparts: RMSNorm versus RMSNorm-Approx, and $\text{BHyT}^*$ versus BHyT. Here, $\text{BHyT}^*$ denotes BHyT with exact variance computation, whereas BHyT uses the proposed variance approximation. All models are evaluated using the same downstream pipeline as in the main experiments.

This ablation verifies whether the approximation improves throughput without materially changing convergence or down-stream behavior. As shown in Table 4, the approximation improves training throughput for both RMSNorm and BHyT while preserving comparable evaluation loss, perplexity, and downstream accuracy.

## C.7. Detailed comparison of normalization baselines

Table 8 clarifies the efficiency–stability trade-off across the baselines. RMSNorm and LNS compute exact statistics twice per block, Peri-LN doubles this cost, DyT eliminates statistics entirely, and BHyT requires only one exact statistic thanks to the variance approximation used for the second sublayer.

*Table 8.* Comparison of the normalization baselines considered in this paper. The table summarizes each method's functional form and the number of exact statistics computed per Transformer block.

| Method | Formulation | # exact stats per block |
|---|---|---|
| RMSNorm | $\gamma \odot (x/s_x)$ | 2 |
| LNS | $(1/\sqrt{\ell})\,\gamma \odot (x/s_x)$ | 2 |
| Peri-LN | RMSNorm before/after each sublayer | 4 |
| DyT | $\gamma \odot \tanh(\alpha_{\mathrm{DyT}}x)$ | 0 |
| BHyT | $\gamma \odot \tanh((\lambda/\kappa)\,x/s_x)$ | 1 |

## C.8. Time to target loss analysis

To assess how efficiently each normalization method reduces loss during pretraining, we measure the time required to reach the loss achieved by BHyT approximately 5,000 steps before its training is completed. For both the Llama-1B and Llama-3B models, we track this target loss and evaluate competing methods at the point when (and if) they reach it. As shown in Table 9, most baselines reach this loss only much later in training or fail to reach it at all within the allotted training budget, indicating substantially slower effective convergence.

*Table 9.* Time-to-target-loss comparison for Llama-1B and Llama-3B. Each method is evaluated at the loss value that BHyT reaches roughly 5K steps before completing pretraining. Most baselines reach this loss only much later or fail to reach it within the training budget.

| | | | | | Llama-1B | | | | | | |
|---|---|---|---|---|---|---|---|---|---|---|---|
| Method | PT Train Loss | Wall time (hour) | Checkpoint Step | Arc-e | PIQA | Hellaswag | OpenBookQA | Winogrande | MMLU | BoolQ | Avg. |
| RMSNorm | 3.282 | 39.35 | 49K | $30.97_{\pm0.20}$ | $62.89_{\pm0.74}$ | $32.77_{\pm0.09}$ | $32.32_{\pm0.94}$ | $\mathbf{50.54}_{\pm0.48}$ | $\mathbf{25.70}_{\pm0.01}$ | $56.26_{\pm0.44}$ | $41.64_{\pm0.10}$ |
| Peri-LN | 3.288 | 42.62 | 50K | $\mathbf{31.63}_{\pm0.32}$ | $\mathbf{63.07}_{\pm0.43}$ | $32.05_{\pm0.17}$ | $32.40_{\pm0.80}$ | $49.41_{\pm1.64}$ | $24.91_{\pm0.00}$ | $58.05_{\pm0.34}$ | $41.65_{\pm0.12}$ |
| LNS | 3.281 | 37.65 | 49K | $31.39_{\pm0.24}$ | $62.94_{\pm0.64}$ | $\mathbf{32.80}_{\pm0.16}$ | $33.50_{\pm0.58}$ | $50.37_{\pm0.91}$ | $24.88_{\pm0.01}$ | $56.67_{\pm0.37}$ | $41.79_{\pm0.26}$ |
| BHyT | 3.289 | $\mathbf{33.22}$ | $\mathbf{46K}$ | $30.50_{\pm0.16}$ | $62.42_{\pm0.41}$ | $32.11_{\pm0.10}$ | $33.88_{\pm0.73}$ | $50.15_{\pm0.34}$ | $25.19_{\pm0.02}$ | $\mathbf{61.86}_{\pm0.17}$ | $\mathbf{42.30}_{\pm0.13}$ |
| | | | | | Llama-3B | | | | | | |
| Method | PT Train Loss | Wall time (hour) | Checkpoint Step | Arc-e | PIQA | Hellaswag | OpenBookQA | Winogrande | MMLU | BoolQ | Avg. |
| RMSNorm | 3.203 | 76.55 | 60K | $31.01_{\pm0.21}$ | $\mathbf{66.57}_{\pm0.41}$ | $\mathbf{41.52}_{\pm0.25}$ | $\mathbf{33.92}_{\pm0.33}$ | $50.50_{\pm0.78}$ | $25.67_{\pm0.02}$ | $37.94_{\pm0.12}$ | $40.89_{\pm0.18}$ |
| Peri-LN | 3.165 | 88.72 | 60K | $31.82_{\pm0.38}$ | $64.52_{\pm0.24}$ | $36.05_{\pm0.11}$ | $32.28_{\pm0.59}$ | $49.30_{\pm0.52}$ | $25.99_{\pm0.00}$ | $57.46_{\pm0.31}$ | $42.49_{\pm0.25}$ |
| LNS | 3.160 | 78.55 | 60K | $\mathbf{31.88}_{\pm0.42}$ | $64.68_{\pm0.46}$ | $36.25_{\pm0.10}$ | $32.45_{\pm0.53}$ | $51.18_{\pm0.75}$ | $\mathbf{26.87}_{\pm0.01}$ | $37.83_{\pm0.08}$ | $41.16_{\pm0.11}$ |
| BHyT | 3.135 | $\mathbf{64.30}$ | $\mathbf{56K}$ | $31.84_{\pm0.15}$ | $65.08_{\pm0.39}$ | $36.48_{\pm0.04}$ | $31.76_{\pm0.33}$ | $\mathbf{51.62}_{\pm0.66}$ | $25.70_{\pm0.01}$ | $60.84_{\pm0.29}$ | $\mathbf{43.33}_{\pm0.16}$ |

Peri-LN, selected as the strongest stability-oriented baseline, eventually approaches the target loss but requires notably more wall-clock time due to the additional normalization applied before and after each sublayer. RMSNorm and LNS converge more slowly and often do not reach the target loss during pretraining. DyT is excluded from this comparison. Although DyT achieves higher training throughput, it remains highly sensitive to hyperparameter choices and does not consistently reduce pretraining loss, making it unsuitable for a meaningful time-to-loss comparison. Overall, BHyT reaches the target loss significantly earlier than all baselines while maintaining competitive downstream accuracy, demonstrating that BHyT provides both stable and efficient optimization at scale.

## C.9. Training stability under varying learning rates: DyT vs. BHyT

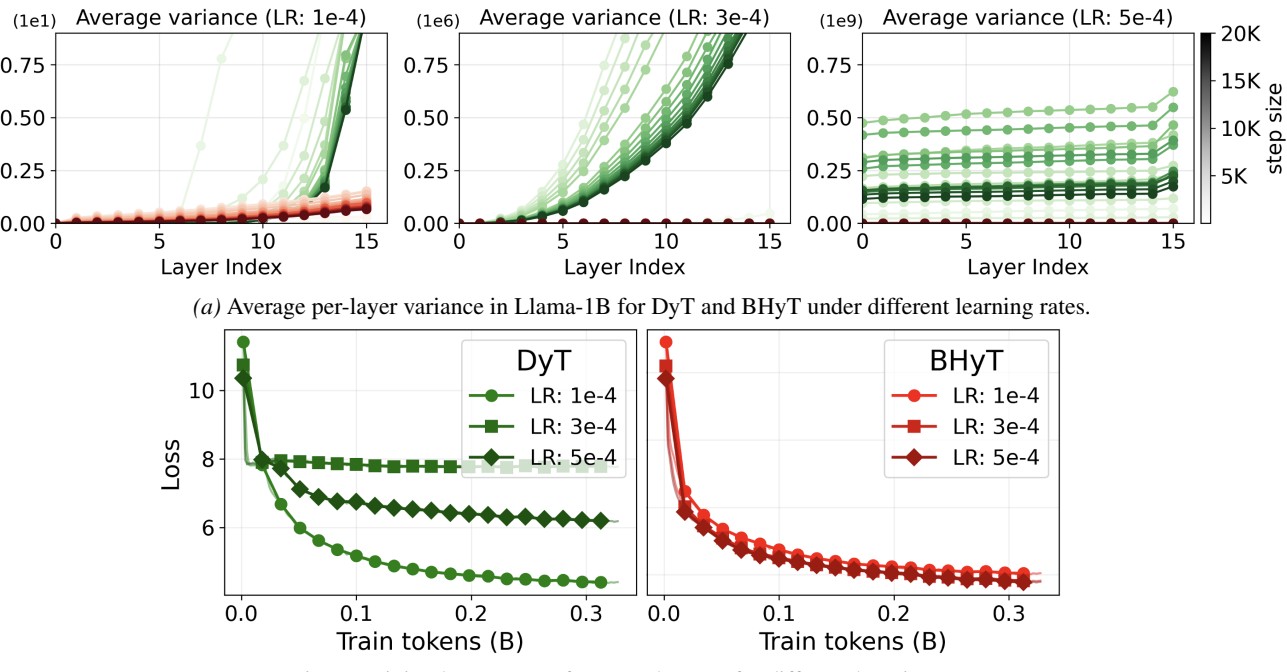

*(a)* Average per-layer variance in Llama-1B for DyT and BHyT under different learning rates.

*(b)* Pretraining loss curves of DyT and BHyT for different learning rates.

*Figure 5.* Learning-rate robustness of DyT and BHyT with tanh-based method in Llama-1B. (a) Average per-layer variance as a function of depth under different learning rates; DyT exhibits rapidly increasing variance at larger learning rates, while BHyT maintains a much smaller variance that grows roughly linearly with depth. (b) Pretraining loss versus training tokens for DyT (left) and BHyT (right) across learning rates, where BHyT converges stably over a wide range of learning rates.

We compare the training stability of DyT and BHyT, which replace the normalization layer with a $\mathrm{tanh}$-based layer when the learning rate is varied. Figure 5 shows that, for DyT, increasing the learning rate leads to a sharp growth in variance amplification, making pretraining harder to optimize. In contrast, BHyT exhibits a much smaller variance scale that grows roughly linearly with depth, and its pretraining converges stably across a wide range of learning rates. These results indicate that BHyT offers robust training stability for large-scale models such as LLMs, enabling more efficient and reliable hyperparameter sweeps.

## C.10. Why BHyT suppresses depth-wise growth more effectively than DyT

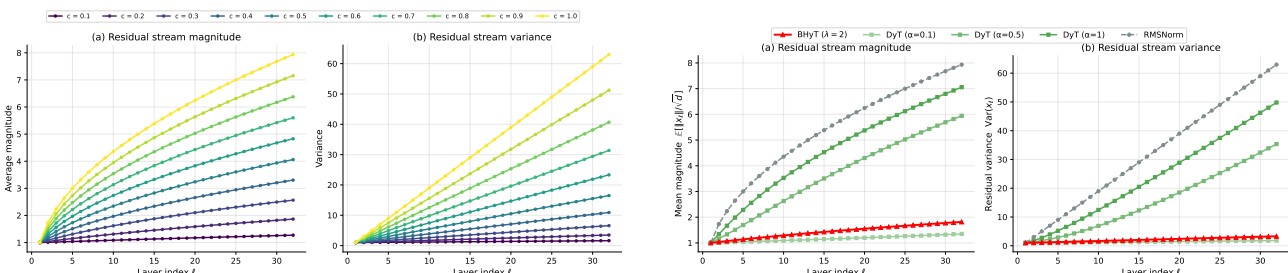

*Figure 6.* Empirical verification of depth-wise residual growth in a 32-layer Pre-LN toy stack ($d = 2048$, $L = 32$). Left: when the per-layer sublayer output is replaced with constant-variance noise, $f(x_\ell) = c\,\boldsymbol{\eta}_\ell$ with $\boldsymbol{\eta}_\ell \sim \mathcal{N}(0, I_d)$, residual magnitude and variance are governed almost entirely by $c$; in this setup, growth becomes rapid once $c \gtrsim 0.7$. Right: under the same toy recursion instantiated with RMSNorm, DyT, and BHyT, DyT remains non-saturating only for sufficiently small $\alpha_{\mathrm{DyT}}$ (roughly $\alpha_{\mathrm{DyT}} \lesssim 0.5$ here), whereas BHyT's adaptive scaling keeps the pre-$\mathrm{tanh}$ argument bounded and yields the flattest depth-wise profile.

We next provide an empirical verification of the claim that depth-wise residual variance growth is governed by the magnitude of the per-layer sublayer output. To isolate this mechanism, we study a 32-layer Pre-LN toy residual stack with hidden size

$d = 2048$ and depth $L = 32$:

$$x_{\ell+1} = x_\ell + f(x_\ell)W_\ell, \qquad W_\ell \sim \mathcal{N}(0, 2/d).$$

This is deliberately not a Transformer block; the goal is to isolate residual-stream evolution as a function of the sublayer map $f(\cdot)$. The corresponding comparison with full Transformer blocks in Llama-1B is given in Figure 3. Figure 6 summarizes the controlled toy-stack comparisons discussed below.

For the left panel of Figure 6, we first set

$$f(x_\ell) = c \cdot \boldsymbol{\eta}_\ell, \qquad \boldsymbol{\eta}_\ell \sim \mathcal{N}(0, I_d),$$

so that the per-layer sublayer output has a directly controlled variance scale. The result shows that residual-stream growth is essentially set by this magnitude: in this toy stack, once $c \gtrsim 0.7$, both residual magnitude and residual variance increase rapidly with depth, whereas smaller $c$ keeps the depth-wise profile comparatively stable.

For the right panel, we instantiate the same toy recursion with RMSNorm, DyT, and BHyT. In this setting, DyT remains in a non-saturating regime only when $\alpha_{\text{DyT}} \lesssim 0.5$; for larger values, the argument of $\tanh$ saturates, the sublayer output becomes order-one, and the residual stack enters the same fast-growth regime seen in the constant-$c$ experiment. This interpretation is consistent with Figure 3: even a small fixed $\alpha_{\text{DyT}}$ does not reliably rescue DyT on Llama-1B, because $\alpha_{\text{DyT}}x$ does not adapt to post-attention statistics and therefore grows with the residual stream into saturation. By contrast, BHyT rescales the pre-$\tanh$ input by $\lambda/(\kappa s_x)$ on each token, keeping the argument bounded by $\lambda$ across depth and yielding the flattest variance profile.

### C.11. Variance approximation

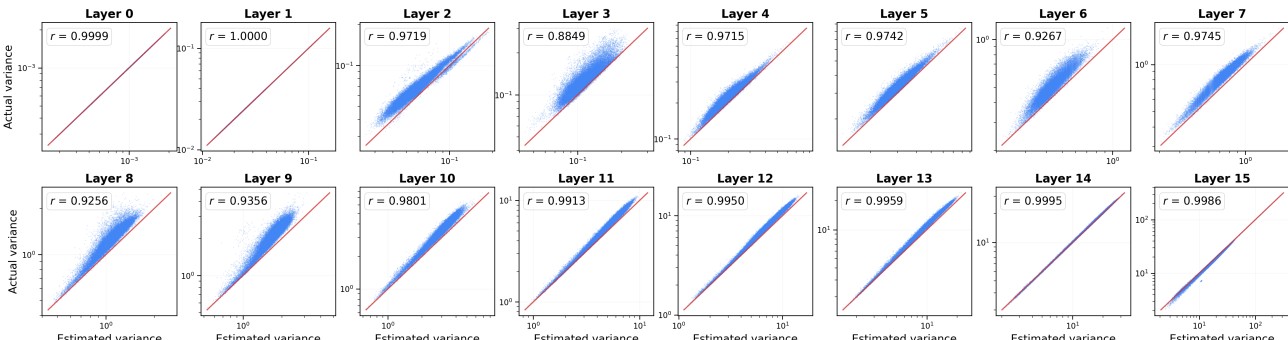

*Figure 7.* Layer-wise variance approximation quality for Llama-1B after pretraining on 20B tokens.

Figure 7 shows the layer-wise variance approximation quality for Llama-1B after pretraining on 20B tokens. The early layers exhibit noticeably improved agreement between the estimated and ground-truth variances.

Because BHyT reuses an approximated variance across subsequent block computations, approximation errors can propagate through the residual stream and accumulate with depth. This makes the earliest layers particularly important: small errors introduced near the input can affect the variance estimates of many later layers. To examine this effect, we also evaluate a hybrid variant that computes the exact variance only in the first two layers and then uses the standard approximation for the remaining layers. This simple correction substantially reduces accumulated approximation error and improves agreement between the estimated and ground-truth variances, indicating that most of the approximation error can be controlled by anchoring the early residual stream with a small number of exact variance computations.

### C.12. Token Generation Speed

We provide additional details for the generation-throughput analysis in Table 5. All measurements are conducted on a single NVIDIA RTX PRO 6000 Blackwell GPU using fp16 autocast. Unless otherwise stated, we use batch size 8, input length 512, greedy decoding, and KV cache. Throughput is computed as the number of generated tokens divided by wall-clock time. For each method, we discard 8 warmup batches and report results over 160 timed batches, obtained from 32 measured batches repeated 5 times.

*Table 10.* Generation throughput across output lengths. Values report mean ± standard deviation over five repeats, and percentages denote relative speed differences with respect to BHyT.

| Throughput (tokens/s) | Max new token length | | |
|---|---|---|---|
| | 128 | 512 | 1024 |
| RMSNorm | $1152.2_{\pm20.9}$ (-2.1%) | $1181.1_{\pm2.9}$ (-1.6%) | $1147.1_{\pm12.7}$ (-1.6%) |
| Peri-LN | $957.5_{\pm4.3}$ (-18.6%) | $984.4_{\pm6.9}$ (-18.0%) | $1015.2_{\pm5.0}$ (-12.9%) |
| LNS | $1095.1_{\pm39.9}$ (-6.9%) | $1150.7_{\pm11.7}$ (-4.1%) | $1140.2_{\pm2.1}$ (-2.2%) |
| DyT | $1363.9_{\pm9.3}$ (+15.9%) | $1352.0_{\pm11.9}$ (+12.7%) | $1269.7_{\pm3.2}$ (+9.0%) |
| BHyT | $1176.7_{\pm39.1}$ | $1199.9_{\pm3.0}$ | $1165.4_{\pm7.7}$ |

*Table 11.* Normalization-only execution time (ms) for Llama-1B across sequence lengths. The upper block reports the standard BHyT implementation with variance approximation, whereas the lower block reports a variant that computes the second variance exactly inside the layer. With the approximation, BHyT remains close to DyT; without it, the exact variance computation becomes a major latency bottleneck.

| With variance approximation (standard BHyT) | | | | | | |
|---|---|---|---|---|---|---|
| Method | 512 | 768 | 1024 | 2048 | 4096 | 8192 |
| RMSNorm | $79.92_{\pm3.35}$ | $79.92_{\pm1.83}$ | $79.62_{\pm1.75}$ | $81.36_{\pm1.38}$ | $85.21_{\pm1.59}$ | $121.63_{\pm0.74}$ |
| Peri-LN | $79.92_{\pm2.36}$ | $80.31_{\pm1.27}$ | $80.08_{\pm1.49}$ | $79.57_{\pm1.66}$ | $84.62_{\pm0.86}$ | $122.15_{\pm0.82}$ |
| LNS | $109.92_{\pm2.41}$ | $109.89_{\pm1.82}$ | $110.05_{\pm2.08}$ | $88.91_{\pm1.61}$ | $92.43_{\pm16.95}$ | $125.03_{\pm1.42}$ |
| DyT | $\mathbf{35.91}_{\pm1.05}$ | $\mathbf{36.43}_{\pm0.87}$ | $\mathbf{36.13}_{\pm0.49}$ | $\mathbf{38.10}_{\pm0.38}$ | $\mathbf{41.38}_{\pm0.80}$ | $\mathbf{61.02}_{\pm1.60}$ |
| BHyT | $40.21_{\pm1.02}$ | $40.12_{\pm0.78}$ | $40.34_{\pm0.57}$ | $42.31_{\pm0.39}$ | $45.94_{\pm0.60}$ | $63.22_{\pm0.38}$ |
| Without variance approximation (exact second-variance computation) | | | | | | |
| Method | 512 | 768 | 1024 | 2048 | 4096 | 8192 |
| RMSNorm | $79.28_{\pm1.27}$ | $79.31_{\pm1.78}$ | $79.42_{\pm1.37}$ | $81.16_{\pm1.33}$ | $84.66_{\pm0.80}$ | $121.73_{\pm0.66}$ |
| Peri-LN | $79.62_{\pm1.31}$ | $79.43_{\pm1.43}$ | $79.47_{\pm1.79}$ | $79.05_{\pm1.28}$ | $84.69_{\pm0.95}$ | $122.10_{\pm0.88}$ |
| LNS | $109.23_{\pm1.97}$ | $109.29_{\pm1.84}$ | $106.97_{\pm7.17}$ | $88.63_{\pm1.69}$ | $90.73_{\pm1.24}$ | $124.94_{\pm0.71}$ |
| DyT | $\mathbf{35.87}_{\pm0.47}$ | $\mathbf{36.44}_{\pm0.44}$ | $\mathbf{35.97}_{\pm0.79}$ | $\mathbf{38.21}_{\pm0.81}$ | $\mathbf{41.43}_{\pm0.93}$ | $\mathbf{62.51}_{\pm0.52}$ |
| BHyT | $82.38_{\pm1.60}$ | $81.71_{\pm1.23}$ | $82.52_{\pm6.04}$ | $85.31_{\pm1.14}$ | $98.04_{\pm1.04}$ | $134.86_{\pm1.26}$ |

Table 10 reports generation throughput across different maximum numbers of generated tokens. BHyT consistently outperforms RMSNorm, LNS, and Peri-LN, showing that its efficiency advantage over normalization-based baselines persists across generation lengths. DyT achieves the highest throughput because it removes statistic computation entirely, but it provides weaker activation control than BHyT, as discussed in Section 4.3.

To isolate the source of BHyT's efficiency gain, we also measure the latency of the normalization or normalization-replacement component alone. This breakdown separates the cost of statistic computation from the rest of the Transformer block. The results show that the proposed variance approximation reduces the overhead of BHyT relative to exact-statistic variants, supporting its role as a practical RMSNorm replacement.

Table 11 reports normalization-only execution time with and without the variance approximation. With the standard approximation, BHyT remains close to DyT across sequence lengths. Without the approximation, exact variance computation inside the layer substantially increases latency, confirming that the approximation is essential to BHyT's efficiency.

