# OpenReview forum: "Bounded Hyperbolic Tangent: A Stable and Efficient Alternative to Pre-Layer Normalization in Large Language Models"
_ICML.cc/2026/Conference — ICML 2026 regular_

### Official Review · Reviewer_fjRQ · 2026-03-11

**Soundness:** 2
**Presentation:** 3
**Significance:** 2
**Originality:** 3
**Overall Recommendation:** 4
**Confidence:** 3

**Summary:**

This paper presents BHyT, which is a plug-and-play replacement for pre-LN.  Prior method like RMSNorm and DyT suffer from problems like the curse of depth.  Built upon DyT, BHyT incorporates data-dependent input bounding to keep activations within a stable range. In addition, BHyT approximate the variance in the second normalization layer to improve the efficiency. It improves both training efficiency and inference throughput, with slight improvement on loss / downstream evals.

**Compliance With Llm Reviewing Policy:**

Affirmed.

**Final Justification:**

The rebuttal addressed my main concerns, and I raised my score.

**Key Questions For Authors:**

My other concern is that the early layer show great discrepancy between the approximated and ground-truth activation variances for the second BHyT layer. This may cause error accumulation across layers. Have you considered selectively applying the approximation? For example, not using approximation for the first two layers and only applying approximation for the rest of the layers. It might improve the downstream performance of BHyT without sacrificing too much of the efficiency gains.

**Limitations:**

yes

**Strengths And Weaknesses:**

The paper articulate everything clearly and is very easy to read. In general I think the experiments / analysis / visualization are comprehensive. BHyT clearly have some advantages over the prior methods, in terms of stability. It is also more robust to the choices of learning rate, compared to DyT.


My biggest concern is that some of the main results (table 1 and table 2) really do not show DyT perform better than other baselines. Even though the average score is higher for BHyT, it is primarily driven by BoolQ. And if you remove BoolQ from the eval suites then BHyT does not show too much advantage over other methods.

The experiments using 20B token is support the argument of the paper the best. BHyT clearly does better than the Peri-LN. But I am not sure if BHyT will be better than other baselines under the 20B token setting. While it is understandable that compute might limit what kinds of experiments you could run, it still remains as a concern.

I also think it would be better to include a wider evaluation sets, for example including arc-c and other math datasets, just to show that the current selection of the evals are not cherry-picked.

---

> ### Author Rebuttal · Authors · 2026-03-31
>
> **W1 & W2. Inconsistent Advantage over Baselines & Unclear Advantage at the 20B-Token Scale**
>
> We appreciate the reviewer’s helpful suggestions and insightful comments.
>
> We agree that the short-budget tables alone do not establish broad superiority over DyT and other baselines, and that the original averages can be influenced by BoolQ. To address this concern directly, we added a 20B-token comparison across all baselines.
>
> To address this concern directly, we trained all methods on Llama-1B for 20 billion tokens during the revision period.
>
> **Llama-1B (Pretrained on 20B tokens only)**
>
> | Method | PT Train Loss | PT Eval Loss | Arc-e | Arc-c | PIQA | Hellaswag | OpenBookQA | Winogrande | MMLU | BoolQ | Avg. |
> |---|---|---|---|---|---|---|---|---|---|---|---|
> | RMSNorm | 2.876 | 2.877 | 43.56% | _25.94%_ | _70.46%_ | 45.56% | 28.40% | 50.12% | 23.02% | _59.91%_ | 43.37% |
> | Peri-LN | 2.884 | 2.884 | 43.86% | 24.40% | 69.80% | 44.82% | 30.40% | 52.33% | _22.97%_ | 59.45% | 43.50% |
> | LNS | **2.848** | **2.848** | **45.96%** | 25.00% | **70.95%** | **47.00%** | _30.60%_ | **53.43%** | _22.97%_ | _59.91%_ | **44.48%** |
> | DyT | 2.891 | 2.894 | 44.28% | 25.17% | 70.13% | 44.41% | 30.40% | 51.30% | 22.95% | **59.94%** | 43.57% |
> | BHyT | **2.848** | _2.850_ | _45.58%_ | **26.11%** | 69.97% | _46.24%_ | **31.80%** | _52.49%_ | **23.54%** | 57.09% | _44.10%_ |
>
> **Llama-1B (Pretrained on 20B tokens & SFT)**
>
> | Method | Arc-e | Arc-c | PIQA | Hellaswag | OpenBookQA | Winogrande | MMLU | BoolQ | Avg. |
> |---|---|---|---|---|---|---|---|---|---|
> | RMSNorm | 45.66% | _27.30%_ | _70.13%_ | 46.10% | 28.20% | 49.72% | 23.03% | **58.13%** | 43.54% |
> | Peri-LN | 46.00% | 25.09% | 69.26% | 45.63% | 30.20% | _52.09%_ | 23.00% | 57.49% | 43.59% |
> | LNS | 45.58% | 27.05% | 68.93% | _47.27%_ | 28.20% | 51.78% | 23.18% | 55.75% | 43.47% |
> | DyT | _46.17%_ | 25.68% | 69.97% | 45.79% | _30.60%_ | 51.22% | _23.27%_ | _57.83%_ | _43.82%_ |
> | BHyT | **46.42%** | **27.39%** | **70.29%** | **47.57%** | **32.80%** | **53.28%** | **24.31%** | 54.98% | **44.63%** |
>
> In the newly added 20B-token Llama-1B comparison, **BHyT is competitive in PT-only (second-best average accuracy, 44.10% vs. 44.48% for LNS) and achieves the best post-SFT average accuracy (44.63%). After LoRA SFT on Lima1k, BHyT improves by 0.53%p on average across 8 downstream tasks; importantly, this gain is not driven by BoolQ, where BHyT is lower than several baselines. These results directly address the reviewer’s concern that the short-budget tables alone were insufficient to establish a broad advantage.**
>
> In the revision, we will add the 20B all-baseline results alongside the existing short-budget tables and revise the framing so that the short-budget results are not presented as the sole evidence for BHyT’s advantage.
>
> **W3. Limited Evaluation Coverage**
>
> We agree that a wider evaluation suite would strengthen the paper. In the new 20B-token experiments, we additionally report ARC-Challenge, where BHyT performs best in both the PT-only and post-SFT settings. For math benchmarks, most standard evaluations are generation-based and require substantially more instruction tuning; we were unable to complete these within the rebuttal period, but will include them in the revised version.
>
> **Q1. Approximation Error in Early Layers and Selective Application**
>
> Thank you very much for this excellent suggestion. We conducted experiments based on your proposal.
>
> Due to time constraints, we used a 374M-size model trained on 1B tokens. Therefore, we compared training loss trends rather than benchmark performance.
>
> We compared three approaches: (1) the original BHyT method, (2) using RMSNorm for the first half of the layers and BHyT for the remaining layers, and (3) computing the exact variance for the first 3 layers and using BHyT's approximation for all subsequent layers.
>
> - [Fig.][`[Llama-374M trained on 1B tokens\] Loss curve (BHyT, RMS+BHyT, BHyT-Exact-3L)`](https://anonymous.4open.science/r/bhyt_icml2026-B390/Llama374M-halfRMS-3L.png)
>
> The results show that computing the exact variance for the first three layers leads to the best convergence.
>
> - [Fig.][`[BHyT-Exact-3L] Per-layer variance approximation scatter plots`](https://anonymous.4open.science/r/bhyt_icml2026-B390/var_approx_scatter_llama374M3L-1B.jpg)
>
> In particular, having accurate variance in the early layers significantly reduces the approximation error in the later layers.
>
> This result suggests that improving early-layer variance accuracy can further stabilize BHyT training, and we will include this discussion in the revised version.

---

> > ### Author Rebuttal · Reviewer_fjRQ · 2026-04-06
> >
> > Thank you for the additional experiments! I think these new experiments strengthen the paper. I am raising my score to 4.

---

### Official Review · Reviewer_x4BF · 2026-03-13

**Soundness:** 3
**Presentation:** 3
**Significance:** 3
**Originality:** 3
**Overall Recommendation:** 4
**Confidence:** 3

**Summary:**

This paper proposes BHyT, a drop-in replacement for Pre-Layer Normalization in Transformers. It bounds tanh inputs using Chebyshev's inequality to prevent activation saturation at depth, and replaces the second per-block normalization with a closed-form variance approximation derived from weight norms. Experiments on Llama-1B/3B show ~15.8% faster training and ~4.2% higher inference throughput vs. RMSNorm, with matching or better downstream accuracy across 7 benchmarks.

**Compliance With Llm Reviewing Policy:**

Affirmed.

**Key Questions For Authors:**

Table 3 only includes Peri-LN at 20B tokens. Can the authors provide RMSNorm results under the same budget? This would clarify whether BHyT's downstream accuracy advantage is robust at a more realistic training scale.

**Limitations:**

Yes

**Strengths And Weaknesses:**

Strengths

The combination of probabilistic input bounding with a parallelizable variance approximation is a novel and well-motivated design; the insight that attention-layer variance can be estimated purely from weight norms is technically interesting.

Experiments are thorough: multiple seeds, two model scales, pretraining + SFT, ablation on variance approximation, and a 20B-token run for scalability.

The hyperparameter appendix provide enough detail for reproducibility.

Theorem 3.5 provides a concrete sufficient condition (λ/κ < 1/√L) under which BHyT has provably smaller variance growth than LNS.

Weaknesses

The abstract claims BHyT's "gradient scale is provably upper-bounded by that of RMSNorm," but Theorem 3.5 only compares against LNS, the RMSNorm stability claim is not formally proven anywhere in the paper.

Assumption 3.3 (uniform attention weights, i.i.d. normal weights/activations) is strong and only loosely validated; the approximation accuracy shown in Figure 2 covers a single checkpoint of Llama-1B, leaving open how well it holds early in training or at larger scales.

The 20B-token experiment (Table 3) only compares BHyT against Peri-LN, omitting RMSNorm, LNS, and DyT, making it hard to assess whether the advantage over the primary baseline (RMSNorm) scales with tokens.

Derf (Chen et al., arXiv 2512.10938, Dec 2025), a concurrent normalization-free method using erf-based activation, is directly relevant but absent from the related work and experiments.

---

> ### Author Rebuttal · Authors · 2026-03-31
>
> **W1. Mismatch Between Abstract Claim and Theoretical Result**
>
> Thank you for this precise observation. Thm 3.5 addresses the comparison with LNS. The gradient bound relative to RMSNorm is based on a separate, deterministic argument, which we will make explicit in the revised version.
>
> The key idea is straightforward. Let $J^{\text{BHyT}}$ and $J^{\text{RMS}}$ be the Jacobians of BHyT and RMSNorm, respectively. By direct computation, $J^{\text{BHyT}} = D \cdot J^{\text{RMS}}$, where $D = \text{diag}(\text{sech}^2(\alpha_i) \cdot \lambda/\kappa)$. Since $\text{sech}^2(\cdot) \leq 1$ always holds, every diagonal entry of $D$ is at most $\lambda/\kappa$. By submultiplicativity of matrix norms:
>
> $$J^{\text{BHyT}}_2 \leq \frac{\lambda}{\kappa} \cdot J^{\text{RMS}}_2$$
>
> This bound is deterministic — it holds for all inputs without any distributional assumptions. With default hyperparameters ($\lambda = 1$, $\kappa = 10$), BHyT's gradient scale is at most $1/10$ of RMSNorm's. This is the basis for the "provably upper-bounded" claim. We will add a full derivation in the appendix.
>
> **W2. Limited Validation of Assumption 3.3**
>
> To address whether BHyT's variance approximation degrades over training, we measured approximation quality at multiple checkpoints (12K, 14K, 15.3K steps) of our 1B model, reporting RMSE, $R^2$, Pearson $r$, and Spearman $\rho$ between actual and approximated post-residual variance (100 C4 validation samples, all 16 layers).
>
> **Table: Aggregated variance approximation quality over training.**
>
> | Step | RMSE (↓) | $R^2$ (↑) | Pearson $r$ (↑) | Spearman $\rho$ (↑) |
> |------|----------|--------|---------------|----------------|
> | 12K  | 0.175    | 0.9998 | 0.9999        | 0.9993         |
> | 14K  | 0.167    | 0.9999 | 0.9999        | 0.9993         |
> | 15.3K| 0.166    | 0.9999 | 1.0000        | 0.9993         |
>
> RMSE, $R^2$, and Pearson $r$ all improve monotonically. Spearman $\rho$ shows a marginal change in the fourth decimal place ($0.99926 \rightarrow 0.99925$). Statistical tests confirm this is not a meaningful degradation:
>
> | Test | Statistic | p-value |
> |------|-----------|---------|
> | Paired t-test | t = 1.10 | 0.288 |
> | Wilcoxon signed-rank | W = 49.0 | 0.348 |
> | Sign test | 11+/5− | 0.210 |
>
> Effect size: Cohen's $d = -0.28$ (small). Bootstrap 95% CI of the mean difference: $[-0.0064, +0.0006]$, which includes zero. The apparent decline is attributable to the first two layers (layers 0–1), where Spearman $\rho$ is inherently lower ($0.53$–$0.66$) due to the difficulty of approximating variance at the embedding-to-attention boundary. 11 out of 16 layers actually improved — this is a composition artifact, not a systematic trend.
>
> Per-layer scatter plots of Llama-1B and Llama-3B (28 layers) are also available in [Fig.][`[Llama-1B, trained 20B] Per-layer variance approximation scatter plots`](https://anonymous.4open.science/r/bhyt_icml2026-B390/var_approx_scatter_llama1B-20B.jpg) and [Fig.][`[Llama-3B, trained 20B] Per-layer variance approximation scatter plots`](https://anonymous.4open.science/r/bhyt_icml2026-B390/var_approx_scatter_llama3B-20B.jpg), showing that the approximation remains accurate even in the deeper Llama-3B model.
>
> **W3 & Q1. Incomplete 20B-Token Baseline Comparison**
>
> We have trained all methods on Llama-1B for 20B tokens.
>
> - [Table][`[Llama-1B, 20B tokens] Benchmark results: Pretraining only vs. Pretraining + SFT`](https://anonymous.4open.science/r/bhyt_icml2026-B390/llama-1B_PT20B_eval.png)
>
> In the newly added 20B-token Llama-1B comparison, BHyT is competitive in PT-only (second-best average accuracy, 44.10% vs. 44.48% for LNS) and achieves the best post-SFT average accuracy (44.63%). After LoRA SFT on Lima1k, BHyT improves by 0.53%p on average across 8 downstream tasks; importantly, this gain is not driven by BoolQ, where BHyT is lower than several baselines.
>
> BHyT achieves the best or second-best pretraining performance and the best average after LoRA fine-tuning with Lima-1K. The original 20B-token results used H100 GPUs (BHyT vs. Peri-LN only); these new experiments cover all baselines on RTX PRO 6000 Blackwell GPUs for consistency, which may cause minor discrepancies. Thanks to your suggestion, we confirmed BHyT's potential at larger training scales and in downstream instruction tuning.
>
> **W4. Missing Discussion of Derf (Chen et al., 2025)**
>
> Thank you for this suggestion. Derf replaces normalization with $\text{erf}(\alpha x + s)$ using freely learnable scalars. The key difference is that Derf's scaling is learned independently of input statistics, whereas BHyT determines $\alpha = \lambda/(\kappa s_x)$ from input variance with a probabilistic non-saturation guarantee, enabling formal depth-wise variance control (Thm 3.5). We also note that Derf's LM experiments are limited to GPT-2 (124M), whereas BHyT shows improvements at the Llama 1B/3B scale with up to 20B tokens. We will include Derf in related works and as a baseline in the revised version.

---

### Official Review · Reviewer_PRvh · 2026-03-13

**Soundness:** 3
**Presentation:** 3
**Significance:** 3
**Originality:** 2
**Overall Recommendation:** 4
**Confidence:** 3

**Summary:**

This paper introduces Bounded Hyperbolic Tangent, a normalization layer to replace traditional normalization layers like LayerNorm and RMSNorm in Transformer architectures. They aim to reduce the computational overhead of calculating statistics (mean/variance) and mitigate the instability problem of deep networks due to uncontrolled variance growth. They conducted experiments with Llama 1b and 3b for all normalization methods.

**Compliance With Llm Reviewing Policy:**

Affirmed.

**Final Justification:**

I maintained my initial positive assessment; the paper provides a reasonable improvement over DyT and new experiments shown in the rebuttal phase also support the claim in the paper.

**Key Questions For Authors:**

NA

**Limitations:**

yes

**Strengths And Weaknesses:**

Strengths:
1. The use of Chebyshev's inequality and variance propagation analysis provides a solid foundation.
2. The variance approximation mechanism is interesting and well-justified, especially for inference.
3. They include extensive ablation studies, hyperparameter sweeps, and comparisons across multiple baselines and model sizes.
4. Experiments on Llama models show that BHyT achieves lower pretraining loss and perplexity, faster inference than normalization baselines.

Weaknesses:
1. The variance approximation relies on strong assumptions (e.g., i.i.d. inputs, uniform attention), which may not hold in all settings.
2. Only tested up to 3B parameters; performance on 10B+ models is not demonstrated. The method introduces additional hyperparameters e.g. lambda that may require tuning. For larger models, it is unclear how these factors will impact the performance.
3. The core idea of using tanh with bounded input is reminiscent of DyT [1], though BHyT adds statistical guarantees and variance approximation.
4. SeedNorm[2] also works on top of DyT, comparing it could enhance the practical value of this work.

[1] Transformers without Normalization, Zhu et al.

[2] SeeDNorm: Self-Rescaled Dynamic Normalization, Cai et al.

---

> ### Author Rebuttal · Authors · 2026-03-31
>
> **W1. Strong Assumptions Behind the Variance Approximation**
>
> Thank you for carefully reviewing the theoretical aspects. We acknowledge that Assumption 3.3 is strong in practice. We validated the approximation quality empirically in Fig. 2, Fig. 6, and Tab. 7, showing that error is larger in early layers but becomes much more accurate in later layers.
>
> During the revision period, we further examined the approximation using a Llama-1B model trained on 20B tokens:
>
> [\[Llama-1B, 20B tokens\] Variance approximation scatter plot](https://anonymous.4open.science/r/bhyt_icml2026-B390/var_approx_scatter_llama1B-20B.jpg)
>
> The early-layer approximation improved noticeably with more training tokens, suggesting that approximation quality scales with corpus size. Across all layers, BHyT's approximation tends to underestimate the actual variance (likely due to terms not captured by the assumptions), but maintains a positive correlation with ground truth and does not cause significant performance issues.
>
> This led us to a hypothesis: improving early-layer accuracy should reduce error in all subsequent layers. We tested this on Llama-374M (1B tokens, C4) by using exact variance for only the first layer:
>
> [\[Llama-374M (exact variance for the 1st layer), 1B tokens\] Variance approximation scatter plot](https://anonymous.4open.science/r/bhyt_icml2026-B390/var_approx_scatter_llama374M-1B.jpg)
>
> The result confirmed our hypothesis — exact variance at just the first layer significantly improves accuracy across all remaining layers. While this introduces a small computational overhead, we believe this is an encouraging direction for future work. We are grateful to the reviewer for helping us discover this and will incorporate these findings in the revised version.
>
> **W2. Unclear Scalability to Larger Models (10B+ models)**
>
> We agree that verifying BHyT on larger models is important for demonstrating scalability, and we consider this a fair point. We faced limitations in computational resources and time for 10B+ pretraining.
>
> That said, competitive baselines were also validated at similar scales: LNS (from 130M to 1B) and Peri-LN (from 400M tO 3.2B). Following these prior works, we chose 1B and 3B as appropriate scales for validating the method itself.
>
> Regarding $\lambda$: as described in Appendix C, we found the best $\lambda$ on Llama-1B (20K steps, validation loss) and applied the same value to Llama-3B. This suggests $\lambda$ is not highly sensitive to model size, and a separate search per scale is likely unnecessary.
>
> **W3. Limited Novelty over DyT**
>
> We acknowledge the surface-level similarity. However, the bounding mechanisms are fundamentally different. DyT's $\alpha$ is a free learnable scalar with no constraints — in deeper layers, $\alpha$ can grow large and cause $\tanh$ saturation, with no theoretical guarantee for stability. BHyT determines $\alpha = \lambda / (\kappa s_x)$ adaptively based on input statistics, and the Chebyshev bound probabilistically guarantees operation in the non-saturating regime. Lemma B.3 provides the theoretical analysis, and Fig. 5 empirically confirms that BHyT is more stable than DyT.
>
> **W4. Missing SeedNorm Comparison**
>
> SeeDNorm (first arXived in October 2025) is concurrent work with BHyT, and we will add it as both a baseline and a related work in the revised version. SeeDNorm is a dynamic normalization method built on RMSNorm: it first normalizes the input and then adjusts the rescaling term using the current input, aiming to preserve input norm information. By contrast, BHyT is a drop-in replacement for Pre-LN that uses bounded tanh with data-driven input scaling and variance control, with the goal of keeping activations in a stable, non-saturated range while also reducing normalization overhead. In short, SeeDNorm extends normalization, while BHyT moves toward a bounded and more efficient normalization-free design.
>
> During the revision period, we conducted a preliminary comparison by pretraining Llama-374M on 1B tokens from C4:
> [\[Llama-374M, 1B tokens\] Training loss curves: BHyT SeedNorm](https://anonymous.4open.science/r/bhyt_icml2026-B390/llama374M-Losscurve.png)
>
> In this experiment, BHyT showed consistently lower training loss than SeeDNorm throughout training. Using SeeDNorm’s reported hyperparameters, SeeDNorm did not match BHyT in this setting. We note, however, that this result is still preliminary, since SeeDNorm was mainly evaluated on OLMo2-style architectures, while our current comparison used Llama-374M. To make the comparison fairer and stronger, we plan to additionally run BHyT on OLMo2-550M and further extend the study to 1B and 3B models trained on 20B tokens in the revised version.

---

> > ### Author Rebuttal · Reviewer_PRvh · 2026-04-02
> >
> > I read the response from the authors. I remain my positive score.

---

### Official Review · Reviewer_482j · 2026-03-23

**Soundness:** 3
**Presentation:** 3
**Significance:** 3
**Originality:** 4
**Overall Recommendation:** 5
**Confidence:** 3

**Summary:**

This paper proposes a novel normalization, BHyT, which can be thought of as a hybrid of RMSNorm and DyT (https://arxiv.org/abs/2503.10622). In essence, BHyT replaces the learned scale vector in RMSNorm with a theory-derived fixed scalar coefficient ($\lambda /\ \kappa $) and then adds a hyperbolic tangent from DyT on top of the RMSNorm. However, it doesn’t completely remove the learned vector scale, rather placing it outside the `tanh` function. What is remarkable about the authors’ approach is that they forgo calculation of variance for 50% of the BHyT instances, replacing it with less compute-expensive approximation. This allows the models augmented with BHyT to become faster than those with plain RMSNorm, as evidenced by token generation up on sequences up to 1024 tokens. The paper also shows that BHyT leads to better modeling performance after pre-training on small token budgets.

**Compliance With Llm Reviewing Policy:**

Affirmed.

**Final Justification:**

Pending on authors expanding upon and incorporating all of the answers to the feedback into their paper, I increase my score and vote for acceptance of their work. Please also answer the comments in "Rebuttal Acknowledgment" and incorporate them, too.

**Key Questions For Authors:**

**1.** I don’t understand how the layer activations magnitudes grow much slower with the increase of layer index in the case of BHyT-augmented Transformer in comparison with DyT? Both normalizations use hyperbolic tangent which limits the inputs to Attention and MLP sub-layers to range from -1 to 1 regardless of calculations inside the `tanh` function.

**2.** Why equation 2 doesn’t include $\mu_x$ as compared to eq. 1? Maybe you have missed writing an assumption that inputs to the BHyT layer should have zero expectation?

**3.** Due to the phrasing on lines 303-316, I’m confused whether you used a token budget of ~2 or 20 billion tokens for your experiments? Specifically, how many tokens were seen by the models in Tables 1-4, 6, 9-11?

**4.**  The throughput gains of BHyT in comparison with RMSNorm (Table 8) are stunning, given that 1) it forgoes calculation of variance only in one half of the Transformer sublayers and introduces other calculations instead, including non-linear $tanh(\odot)$ transformation; 2) the proportion of time to calculate normalizations and other non-linear/elementwise operations in Transformer is relatively small in comparison with raw matrix multiplications. With this in mind, could you please report the exact architecture of the models in Table 8 (number of parameters, layers, hidden dimension $d$, number of heads, MLP expansion ration, etc.)? And could you also provide the speed measurements for the normalizations *only* (not for the end-to-end forward pass of the whole model) with different $d$ ?

**Limitations:**

Yes.

**Strengths And Weaknesses:**

**Strengths**

* I can attest that the problem of computation overhead and inefficiency of standard LayerNorm and RMSNorm is real but it’s often overlooked by the majority of work on improving Transformer architecture and its alternatives. So, this research direction is original and relevant.

* The paper provides an impressive result of the novel normalization being up to 5.9% (53.5/50.5) faster than RMSNorm, judging by Table 8, during autoregressive inference.

* Based on Figures 3-4, the authors successfully accomplish the stated major goal of curbing activation variance with the increase of layer number in deep Transformer architectures, and their method is the best among all considered counterparts.

* Transformer models of size 1-3B equipped with proposed BHyT normalization achieve strong performance at language understanding and generation benchmarks after pre-training on 2B tokens.

* The theoretical estimate of the variance of the input to the MLP sublayers (eq. 4) fits well with the empirical data (Figure 2), despite loose assumptions.

**Weaknesses**


**1.** From the paper, it seems that only Table 11 showcases the results with the token budget 20B while all other validation is performed on models trained on less than 2B tokens (please correct me if I’m wrong). And the results in Table 11 don’t include some of the alternative solutions, including the baseline transformer variant with RMSNorm. From personal experience, I can tell that 2B tokens might be insufficient to judge the modeling performance of architectural modification. A modification with best results at 2B tokens can fall behind significantly at longer training durations. I appreciate that your resources might be constrained and I believe the provided data are enough to tentatively prove the viability of BHyT. But the potential of the novel element could be demonstrated better if you would train your model and all considered alternatives for 20-50B tokens, perhaps at the expense of model size and/ or repeated runs with different seeds (360-440M or even 140-180M parameter range would be enough).

**2.** I'm a strong proponent of making narration self-consistent, however, I missed detailed descriptions of the alternatives for the proposed BHyT layer (DyT, LNS, Peri-LN). Also, the paper’s main objective is to solve the problem of increasing activation variance in deeper layers of a Transformer model, and it equates training stability with low and constant variance of activations. But it doesn’t immediately provide a rigorous explanation and empirical evidence why the growth of activations variance/ magnitude is detrimental, referring instead to the LNS paper. It would benefit the paper if you could include a detailed section in the Appendix detailing the findings of the related works, including the formulas of alternative normalization layers you compare with.

**3.** Assumption 3.3 on lines 205-212 does not necessarily hold in real environments. While with the evolution of weight distribution during pre-training, the matrices $W_O$ and $W_V$ can still keep the approximately normal form (although with fat tails and excess kurtosis), softmax attention weights are definitively non-uniform. The majority of the attention weights for a given query tend to be near zero, but a few tokens hold concentrated probability mass, which significantly diminishes the entropy of attention distribution in comparison with uniform distribution.

**4.** Typo in Figures 3 and 5a: I believe you meant “number of steps” instead of “step size” as the right vertical axis name on the graphs.

---

> ### Author Rebuttal · Authors · 2026-03-31
>
> **W1. Limited Long-Token Training Validation**
>
> We sincerely thank the reviewer for the thoughtful and constructive feedback. We agree and have trained all methods on Llama-1B for 20 billion tokens.
>
> - [Table] [`[Llama-1B, 20B tokens\] Benchmark results: Pretraining only vs. Pretraining + SFT`](https://anonymous.4open.science/r/bhyt_icml2026-B390/llama-1B_PT20B_eval.png)
>
> BHyT is competitive in PT-only (second-best avg. accuracy, 44.10% vs. 44.48% for LNS) and achieves the best post-SFT average (44.63%), improving by 0.53%p on average across 8 downstream tasks. This gain is broad — not driven by any single benchmark.
>
> Our manuscript includes 20B-token experiments in Tab. 3 (Llama-3B) and Tab. 11 (Llama-1B), but both only compare against Peri-LN. We agree that all baselines were needed at this scale. Note that these new experiments used RTX PRO 6000 Blackwell GPUs (original results used H100), which may cause minor discrepancies.
>
> **W2. Insufficient Explanation of Alternatives and Variance Growth**
>
> We thank the reviewer for this helpful comment. We agree that the causal link should be more explicit.
>
> The core issue is: as residual variance grows with depth, the normalized branch becomes increasingly small relative to the residual stream, so each block approaches a near-identity map and the effective use of depth is reduced. This is the "curse of depth" (Sun et al., 2025).
>
> More concretely, Lemma B.4 shows that variance increases monotonically with depth when $C_{\text{Attn}}, C_{\text{MLP}} > 0$ and $\rho_1, \rho_2 \geq 0$ (consistently observed in our measurements), causing $x_{\ell+1} \approx x_\ell + \varepsilon$. Figure 3 confirms this empirically: RMSNorm and DyT show sharp variance increases, whereas BHyT maintains the lowest and most stable profile.
>
> In the revised version, we will make the connection from Lemma B.4 to the near-identity consequence explicit. (Theorem 3.5 and Lemma B.5 already provide formal comparisons in the appendix.)
>
> **W3. Strong Assumptions Behind the Variance Approximation**
>
> We acknowledge that Assumption 3.3 is strong in practice. We validated the approximation quality in Fig. 2, Fig. 6, and Tab. 7, showing that error is larger in early layers but much more accurate in later layers.
>
> We further examined this using a Llama-1B model trained on 20B tokens:
>
> - [Fig.][`[Llama-1B, 20B tokens\] Variance approximation scatter plot`](https://anonymous.4open.science/r/bhyt_icml2026-B390/var_approx_scatter_llama1B-20B.jpg)
>
> Early-layer approximation improved noticeably with more tokens, suggesting quality scales with corpus size. BHyT tends to underestimate actual variance, but maintains a positive correlation with ground truth without causing performance issues.
>
> This led to a hypothesis: improving early-layer accuracy should reduce error in subsequent layers. We tested this on Llama-374M (1B tokens, C4) using exact variance for only the first layer:
>
> - [Fig.] [`[Llama-374M (exact var. for the 1st layer), 1B tokens] Variance approximation scatter`](https://anonymous.4open.science/r/bhyt_icml2026-B390/var_approx_scatter_llama374M-1B.jpg)
>
> Exact variance at just the first layer significantly improves accuracy across all remaining layers. We will incorporate this finding in the revised version.
>
> **W4. Typo in Fig. 3 and 5a** Thank you. We will fix these.
>
> **Q1. Why BHyT Controls Activation Growth Better than DyT**
>
> Even though both BHyT and DyT bound sublayer input to $[-1, 1]$ via $\tanh()$, the residual stream $x_\ell$ remains unbounded. BHyT adaptively scales the input to prevent saturation, whereas DyT's $\alpha$ is a free learnable parameter with no constraints. BHyT therefore achieves less activation growth (Appendix D.2).
>
> **Q2. Missing Zero-Mean Assumption in Eq. 2**
> We assume zero mean following RMSNorm (stated in line 189). We will add an explicit note that $\mu_x = 0$.
>
> **Q3. Confusing Token Budget Description**
>
> We apologize for the confusion. Tables 1, 2, 4, 6, 9, and 10 use  $\sim$ 2B tokens, while Tables 3 and 11 use $\sim$ 20B tokens.
>
> **Q4. Need for Detailed Throughput Setup and Norm-Only Benchmark**
>
> Llama-1B follows the Llama 3.2 1B architecture ($\sim$1.2B params, 16 layers, hidden dim 2048, 32 heads, MLP expansion ratio 4).
>
> BHyT pre-computes the approximated variance outside the normalization layer. We report execution time (ms) under two options:
>
> - [Table][`[Llama-1B\] Normalization-only execution time (ms): with variance approximation`](https://anonymous.4open.science/r/bhyt_icml2026-B390/norm_speed_option1.png)
>
> - [Table][`[Llama-1B\] Normalization-only execution time (ms): without variance approximation`](https://anonymous.4open.science/r/bhyt_icml2026-B390/norm_speed_option2.jpg)
>
> With variance approximation, BHyT is nearly as fast as DyT. Without it, the exact computation plus $\tanh()$ makes it significantly slower — this is why variance approximation is essential to BHyT's design.

---

> > ### Author Rebuttal · Reviewer_482j · 2026-04-04
> >
> > My feedback has mostly been resolved. I have a few remaining questions and I would like the authors to answer them. I’m raising my score to 5.
> >
> > W2: You didn’t address this point: “I missed detailed descriptions of the alternatives for the proposed BHyT layer (DyT, LNS, Peri-LN)” (related work). Could you add that section with detailed explanations/ comparisons of the baseline methods?
> >
> > Q1: Sorry, still couldn’t get it. BHyT also has a “free learnable parameter with no constraints” outside of `tanh`, called $\gamma$. Please explain why BHyT experiences lesser variance growth than DyT in more detail, ideally by comparing the formulations of the two activations in question side-by-side.
> >
> > New question: did the validation experiments measuring modeling quality use BHyT variant with exact or approximate variance calculations?

---

> > > ### Author Response · Authors · 2026-04-07
> > >
> > > We thank you for your continued interest and in-depth follow-up questions.
> > >
> > > **W2. Detailed descriptions of baselines**
> > >
> > > We provide a side-by-side comparison of baseline methods (to be added to the revised manuscript).
> > >
> > > **[Tab.] Normalization layer comparison.**
> > >
> > > | Method | Formulation | # statistics per block |
> > > |---|---|---|
> > > | RMSNorm | $\gamma \odot \dfrac{x}{s_x}$ | 2 |
> > > | LNS | $\dfrac{1}{\sqrt{\ell}} \cdot \gamma \odot \dfrac{x}{s_x}$ | 2 |
> > > | Peri-LN | RMSNorm applied before **and** after each sublayer | 4 |
> > > | DyT | $\gamma \odot \tanh(\alpha_{\text{DyT}} \cdot x)$ | 0 |
> > > | BHyT | $\gamma \odot \tanh\!\left(\dfrac{\lambda}{\kappa} \cdot \dfrac{x}{s_x}\right)$ | 1 |
> > >
> > > **RMSNorm** fixes each sublayer's output variance to $\gamma^2$ but does not suppress residual stream accumulation, leaving the curse of depth unresolved. **LNS** scales RMSNorm by $1/\sqrt{\ell}$ to curb depth-wise growth, but still pays the full per-layer cost. **Peri-LN** normalizes before and after every sublayer (4 reductions) for the strongest control. **DyT** eliminates reductions via $\tanh(\alpha_{\text{DyT}} x)$, but applies $\alpha_{\text{DyT}}$ to the raw input; $\tanh$ saturates as the residual stream grows (analyzed in Q1). **BHyT** uses a learnable $\lambda$ in front of $\tanh$, applied to $x/s_x$ (unit variance by construction), so the pre-$\tanh$ argument stays within $[-\lambda, \lambda]$ with probability $\geq 1 - 1/\kappa^2$ (Prop. 3.1) independent of depth. For efficiency, $s_x$ is computed once per block and the second variance is approximated via Theorem 3.4.
> > >
> > > **Q1. BHyT vs DyT - why BHyT experiences lesser variance growth despite both having $\gamma$**
> > >
> > > $\gamma$ is indeed a free learnable parameter in BHyT, but this does not equalize the two methods. In Pre-LN, every block contributes to the residual stream, so the per-layer sublayer output $f(x_\ell)$ governs how fast residual variance grows. **BHyT structurally keeps this contribution small at every depth; DyT does not.** $\gamma$, sitting outside $\tanh$, only rescales the already-fixed output and cannot affect this accumulation.
> > >
> > > - **Side-by-side formulation.** According to W2 above, the structural difference between DyT's $\alpha_{\text{DyT}} x$ and BHyT's $\lambda x / (\kappa s_x)$ lies inside $\tanh$: BHyT's pre-$\tanh$ argument is normalized by the per-token std, so by Prop. 3.1 it stays within $[-\lambda, \lambda]$ with probability $\geq 99\%$ at every depth, regardless of how the residual stream evolves.
> > >
> > > - **The role of $\gamma$.** RMSNorm has $\gamma$ yet still suffers the curse of depth, and LNS needs an *additional* layer-index factor beyond its $\gamma$ precisely because $\gamma$ alone cannot suppress variance growth. Sitting outside $\tanh$, $\gamma$ only rescales its output: whether that stays small (BHyT) or grows (DyT) is decided inside $\tanh$, before $\gamma$ applies. No $\gamma$ can rescue DyT's saturating sublayer.
> > >
> > > - **Empirical verification.** The claim is that residual variance grows with depth at a rate set by the per-layer sublayer output. We test this on a 32-layer Pre-LN toy stack ($d=2048$, $L=32$): $x_{\ell+1} = x_\ell+f(x_\ell)W_\ell,\;W_\ell\sim\mathcal{N}(0, 2/d).$ Not a transformer; residual evolution depends only on $f$. The Llama-1B comparison with full transformer blocks is in Fig. 3.
> > >
> > >   - First, $f(x_\ell)=c \cdot \boldsymbol{\eta}\_\ell$ with $\boldsymbol{\eta}\_\ell \sim \mathcal N(0, I\_d)$: [[Fig.] Residual stream growth driven by a constant-variance sublayer noise.](https://anonymous.4open.science/r/bhyt_icml2026-B390/fig_constant_magnitude.pdf)
> > >
> > >     Residual stream growth is fully set by the per-layer sublayer output magnitude: once $c > 0.7$, residual variance grows quickly, while small $c$ keeps it stable.
> > >
> > >   - Next, with RMSNorm, DyT and BHyT: [[Fig.] Per-layer residual stream magnitude and variance across normalizations.](https://anonymous.4open.science/r/bhyt_icml2026-B390/fig_depthwise_growth.pdf)
> > >
> > >     DyT stays non-saturating only for $\alpha_{\text{DyT}} \lesssim 0.5$; otherwise $\tanh$ saturates, the sublayer reaches $\sim\!1$, and residual variance accumulates rapidly (matching the constant-$c$ regime). Even small $\alpha_{\text{DyT}}$ does not rescue DyT on Llama-1B (Fig. 3): $\alpha_{\text{DyT}} x$ does not adapt to post-attention statistics, so it grows with the residual stream into saturation. BHyT avoids this because $\lambda/(\kappa s_x)$ adapts to per-token statistics, keeping the $\tanh$ argument bounded by $\lambda$ at every depth.
> > >
> > > **New Question: Exact vs approximate variance in validation experiments**
> > >
> > > This is precisely what ``Sec. 4.6 and Tab. 4 address``: BHyT (approximate) matches BHyT$^*$ (exact) on pretraining loss (3.268 vs. 3.266) and 7-task accuracy (42.30% vs. 42.71%) - well within seed variance - ``while training ~14.9% faster``. The approximation therefore retains full modeling quality at substantially higher throughput, and We will also revise Sec. 4.2 to state this convention explicitly.

---

### Decision · Program_Chairs · 2026-04-30

**Decision:**

Accept (regular)

**Comment:**

The paper proposes BHyT, an efficient alternative to RMSNorm/Pre-LN-style normalization in transformers, mainly motivated as improving over DyT-style methods that bypass normalization with nonlinearity like tanh. Its two core ideas are (1) a high-probability bound on the range of input to tanh, and (2) approximation to roughly halve the compute needed for obtaining the statistics. Empirically, BHyT is competitive with RMSNorm-based baselines while offering better throughput, though the evidence is still somewhat limited by evaluation scope and scale. The reviews are generally favorable.